# Unified Convergence Theory of Stochastic and Variance-Reduced Cubic Newton Methods

## Abstract

We study stochastic Cubic Newton methods for solving general possibly non-convex minimization problems. We propose a new framework, which we call the *helper framework*, that provides a unified view of the stochastic and variance-reduced second-order algorithms equipped with global complexity guarantees. It can also be applied to learning with auxiliary information. Our helper framework offers the algorithm designer high flexibility for constructing and analysis of the stochastic Cubic Newton methods, allowing arbitrary size batches, and the use of noisy and possibly biased estimates of the gradients and Hessians, incorporating both the variance reduction and the lazy Hessian updates. We recover the best-known complexities for the stochastic and variance-reduced Cubic Newton, under weak assumptions on the noise and avoiding artificial logarithms. A direct consequence of our theory is the new lazy stochastic second-order method, which significantly improves the arithmetic complexity for large dimension problems. We also establish complexity bounds for the classes of gradient-dominated objectives, that include convex and strongly convex problems. For Auxiliary Learning, we show that using a helper (auxiliary function) can outperform training alone if a given similarity measure is small.

## 1 Introduction

In many fields of machine learning, it is common to optimize a function $f(\boldsymbol{x})$ that can be expressed as a finite sum:

$$\min_{\boldsymbol{x} \in \mathbb{R}^d} \left\{ f(\boldsymbol{x}) = \frac{1}{n} \sum_{i=1}^{n} f_i(\boldsymbol{x}) \right\}, \tag{1}$$

or, more generally, as an expectation over some given probability distribution: $f(\boldsymbol{x}) = \mathbb{E}_\zeta \big[ f(\boldsymbol{x}, \zeta) \big]$. When $f$ is non-convex, this problem is especially difficult, since finding a global minimum is NP-hard in general [14]. Hence, the reasonable goal is to look for approximate solutions. The most prominent family of algorithms for solving large-scale problems of the form (1) are the *first-order methods*, such as the Stochastic Gradient Descent (SGD) [25, 16]. They employ only stochastic gradient information about the objective $f(\boldsymbol{x})$ and guarantee the convergence to a stationary point, which is a point with a small gradient norm.

Nevertheless, when the objective function is non-convex, a stationary point may be a saddle point or even a local maximum, which is not desirable. Another common issue is that first-order methods typically have a slow convergence rate, particularly when the problem is *ill-conditioned*. Therefore, they may not be suitable when high precision for the solution is required.

To address these challenges, we can take into account *second-order information* (the Hessian matrix) and apply Newton's method (see, e.g. [19]). Among the many versions of this algorithm, the Cubic Newton method [20] is one of the most theoretically established. With the Cubic Newton method, we

can guarantee *global convergence* to an approximate *second-order* stationary point (in contrast, the pure Newton method without regularization can even diverge when it starts far from a neighborhood of the solution). For a comprehensive historical overview of the different variants of Newton's method, see [24]. Additionally, the rate of convergence of the Cubic Newton is *provably better* than those for the first-order methods.

Therefore, theoretical guarantees of the Cubic Newton method seem to be very appealing for practical applications. However, the basic version of the Cubic Newton requires the exact gradient and Hessian information in each step, which can be very expensive to compute in the large scale setting. To overcome this issue, several techniques have been proposed:

- One popular approach is to use inexact *stochastic gradient and Hessian estimates* with sub-sampling [33, 17, 32, 21, 12, 6, 1]. This technique avoids using the full oracle information, but typically it has a slower convergence rate compared to the exact Cubic Newton.

- *Variance reduction* techniques [35, 29] combine the advantages of stochastic and exact methods, achievieng an improved rates by recomputing the full gradient and Hessian information at some iterations.

- *Lazy Hessian* updates [26, 9] utilize a simple idea of reusing an old Hessian for several iterations of a second-order scheme. Indeed, since the cost of computing one Hessian is usually much more expensive than one gradient, it can improve the arithmetic complexity of our methods.

- In addition, exploiting the special structure of the function $f$ (if known) can also be helpful. For instance, some studies [20, 18] consider *gradient-dominated objectives*, a subclass of non-convex functions that have improved convergence rates and can even be shown to converge to the global minimum. Examples of such objectives include convex and star-convex functions, uniformly convex functions, and functions satisfying the PL condition [23] as a special case.

In this work, we revise the current state-of-the-art convergence theory for the stochastic Cubic Newton method and propose a unified and improved complexity guarantees for different versions of the method, which combine all the advanced techniques listed above.

Our developments are based on the new *helper framework* for the second-order optimization, that we present in Section 3. For the first-order optimization, a similar in-spirit techniques called *learning with auxiliary information* was developed recently in [7, 30]. Thus, our results can also be seen as a generalization of the Auxiliary Learning paradigm to the second-order optimization. However, note that in our second-order case, we have more freedom for choosing the "helper functions" (namely, we use one for the gradients and one for the Hessians). That brings more flexibility into our methods and it allows, for example, to use the lazy Hessian updates.

Our new helper framework provides us with a unified view of the stochastic and variance-reduced methods and can be used by an algorithm designed to construct new methods. Thus, we show how to recover already known versions of the stochastic Cubic Newton with the best convergence rates, as well as present the new *Lazy Stochastic Second-Order Method*, which significantly improves the total arithmetic complexity for large-dimension problems.

**Contributions.**

- We introduce the *helper framework* which we argue encompasses multiple methods in a unified way. Such methods include stochastic methods, variance reduction, Lazy methods, core sets, and semi-supervised learning.
- This framework covers previous versions of the variance-reduced stochastic Cubic Newton methods with known rates. Moreover, it provides us with new algorithms that employ *Lazy Hessian* updates and significantly improves the arithmetic complexity (for high dimensions), by using the same Hessian snapshot for several steps of the method.
- In the case of Auxiliary learning we provably show a benefit from using auxiliary tasks as helpers in our framework. In particular, we can replace the smoothness constant by a similarity constant which might be smaller.
- Moreover, our analysis works both for the general class of non-convex functions, as well as for the class of gradient-dominated problems, that includes convex and uniformly convex functions. Hence, in particular, we are the first to establish the convergence rates of the

stochastic Cubic Newton algorithms with variance reduction for the gradient-dominated case.

## 2 Notation and Assumptions

For simplicity, we consider the finite-sum optimization problem (1), while it can be also possible to generalize our results to arbitrary expectations. We assume that our objective $f$ is bounded from below and denote $f^\star := \inf_{\boldsymbol{x}} f(\boldsymbol{x})$, and use the following notation: $F_0 := f(\boldsymbol{x}_0) - f^\star$, for some initial $\boldsymbol{x}_0 \in \mathbb{R}^d$. We denote by $\|\boldsymbol{x}\| := \langle \boldsymbol{x}, \boldsymbol{x} \rangle^{1/2}$, $\boldsymbol{x} \in \mathbb{R}^d$, the standard Euclidean norm for vectors, and the spectral norm for symmetric matrices, $\|\boldsymbol{H}\| := \max\{\lambda_{\max}(\boldsymbol{H}), -\lambda_{\min}(\boldsymbol{H})\}$, where $\boldsymbol{H} = \boldsymbol{H}^\top \in \mathbb{R}^{d \times d}$. We will also use $x \wedge y$ to denote $\min(x, y)$.

Throughout this work, we make the following smothness assumption on the objective $f$ :

---

**Assumption 1 (Lipschitz Hessian)** *The Hessian of $f$ is Lipschitz continuous, for some $L > 0$:*
$$\|\nabla^2 f(\boldsymbol{x}) - \nabla^2 f(\boldsymbol{y})\| \quad \leq \quad L\|\boldsymbol{x} - \boldsymbol{y}\|, \qquad \forall \boldsymbol{x}, \boldsymbol{y} \in \mathbb{R}^d$$

---

Our goal is to explore the potential of using the Cubically regularized Newton methods to solve problem (1). At each iteration, being at a point $\boldsymbol{x} \in \mathbb{R}^d$, we compute the next point $\boldsymbol{x}^+$ by solving the subproblem of the form

$$\boldsymbol{x}^+ \in \operatorname*{arg\,min}_{\boldsymbol{y} \in \mathbb{R}^d} \Big\{ \Omega_{M, \boldsymbol{g}, \boldsymbol{H}}(\boldsymbol{y}, \boldsymbol{x}) := \langle \boldsymbol{g}, \boldsymbol{y} - \boldsymbol{x} \rangle + \tfrac{1}{2} \langle \boldsymbol{H}(\boldsymbol{y} - \boldsymbol{x}), \boldsymbol{y} - \boldsymbol{x} \rangle + \tfrac{M}{6} \|\boldsymbol{y} - \boldsymbol{x}\|^3 \Big\}. \quad (2)$$

Here, $\boldsymbol{g}$ and $\boldsymbol{H}$ are estimates of the gradient $\nabla f(\boldsymbol{x})$ and the Hessian $\nabla^2 f(\boldsymbol{x})$, respectively. Note that solving (2) can be done efficiently even for non-convex problems (see [8, 20, 5]). Generally, the cost of computing $\boldsymbol{x}^+$ is $\mathcal{O}(d^3)$ arithmetic operations, which are needed for evaluating an appropriate factorization of $\boldsymbol{H}$. Hence, it is of a similar order as the cost of the classical Newton's step.

We will be interested to find a second-order stationary point to (1). We call $(\varepsilon, c)$-*approximate second-order local minimum* a point $\boldsymbol{x}$ that satisfies:

$$\|\nabla f(\boldsymbol{x})\| \quad \leq \quad \varepsilon \qquad \text{and} \qquad \lambda_{min}(\nabla^2 f(\boldsymbol{x})) \quad \geq \quad -c\sqrt{\varepsilon},$$

where $\varepsilon, c > 0$ are given tolerance parameters. Let us define the following accuracy measure (see [20]):

$$\mu_c(\boldsymbol{x}) \quad := \quad \max\Big(\|\nabla f(\boldsymbol{x})\|^{3/2}, \tfrac{-\lambda_{min}(\nabla^2 f(\boldsymbol{x}))^3}{c^{3/2}}\Big), \qquad \boldsymbol{x} \in \mathbb{R}^d, \ c > 0.$$

Note that this definition implies that if $\mu_c(\boldsymbol{x}) \leq \varepsilon^{3/2}$ then $\boldsymbol{x}$ is an $(\varepsilon, c)$-approximate local minimum.

**Computing gradients and Hessians.** It is clear that computing the Hessian matrix can be much more expensive than computing the gradient vector. We denote the corresponding arithmetic complexities by *HessCost* and *GradCost*. We will make and follow the convention that $HessCost = d \times GradCost$, where $d$ is the dimension of the problem. For example, this is known to hold for neural networks using the backpropagation algorithm [15]. However, if the Hessian has a sparse structure, the cost of computing the Hessian can be cheaper [22]. Then, we can replace $d$ with the *effective dimension* $d_{\text{eff}} := \frac{HessCost}{GradCost} \leq d$.

## 3 Second-Order Optimization with Helper Functions

In this section, we extend the helper framework previously introduced in [7] for first-order optimization methods to second-order optimization.

**General principle.** The general idea is the following: imagine that, besides the objective function $f$ we have access to a help function $h$ that we think is similar in some sense (that will be defined later) to $f$ and thus it should help to minimize it.

Note that many optimization algorithms can be framed in the following sequential way. For a current state $\boldsymbol{x}$, we compute the next state $\boldsymbol{x}^+$ as:

$$\boldsymbol{x}^+ \quad \in \quad \arg\min_{\boldsymbol{y}\in\mathbb{R}^d}\Big\{\, \hat{f}_{\boldsymbol{x}}(\boldsymbol{y}) + Mr_{\boldsymbol{x}}(\boldsymbol{y}) \,\Big\},$$

where $\hat{f}_{\boldsymbol{x}}(\cdot)$ is an approximation of $f$ around current point $\boldsymbol{x}$, and $r_{\boldsymbol{x}}(\boldsymbol{y})$ is a regularizer that encodes how accurate the approximation is, and $M > 0$ is a regularization parameter. In this work, we are interested in cubically regularized second-order models of the form (2) and we use $r_{\boldsymbol{x}}(\boldsymbol{y}) := \frac{1}{6}\|\boldsymbol{y} - \boldsymbol{x}\|^3$.

Now let us look at how we can use a helper $h$ to construct the approximation $\hat{f}$. We notice that we can write

$$f(\boldsymbol{y}) \quad := \quad \underbrace{h(\boldsymbol{y})}_{\text{cheap}} + \underbrace{f(\boldsymbol{y}) - h(\boldsymbol{y})}_{\text{expensive}}$$

We discuss the actual practical choices of the helper function $h$ below. We assume now that we can afford the second-order approximation for the cheap part $h$ around the current point $\boldsymbol{x}$. However, approximating the part $f - h$ can be expensive (as for example when the number of elements $n$ in finite sum (1) is huge), or even impossible (due to lack of data). Thus, we would prefer to approximate the expensive part less frequently. For this reason, let us introduce an extra *snapshot point* $\hat{\boldsymbol{x}}$ that is updated less often than $\boldsymbol{x}$. Then, we use it to approximate $f - h$. Another question that we still need to ask is *what order should we use for the approximation of $f - h$?* We will see that order 0 (approximating by a constant) leads as to the basic stochastic methods, while for orders 1 and 2 we equip our methods with the variance reduction.

Combining the two approximations for $h$ and $f - h$ we get the following model of our objective $f$:

$$\hat{f}_{\boldsymbol{x},\tilde{\boldsymbol{x}}}(\boldsymbol{y}) \quad = \quad C(\boldsymbol{x},\tilde{\boldsymbol{x}}) + \langle \mathcal{G}(h,\boldsymbol{x},\tilde{\boldsymbol{x}}), \boldsymbol{y} - \boldsymbol{x}\rangle + \tfrac{1}{2}\langle \mathcal{H}(h,\boldsymbol{x},\tilde{\boldsymbol{x}})(\boldsymbol{y}-\boldsymbol{x}), \boldsymbol{y}-\boldsymbol{x}\rangle, \tag{3}$$

where $C(\boldsymbol{x},\tilde{\boldsymbol{x}})$ is a constant, $\mathcal{G}(h,\boldsymbol{x},\tilde{\boldsymbol{x}})$ is a linear term, and $\mathcal{H}(h,\boldsymbol{x},\tilde{\boldsymbol{x}})$ is a matrix. Note that if $\tilde{\boldsymbol{x}} \equiv \boldsymbol{x}$, then the best second-order model of the form (3) is the Taylor polynomial of degree two for $f$ around $\boldsymbol{x}$, and that would give us the exact Newton-type method. However, when the points $\boldsymbol{x}$ and $\tilde{\boldsymbol{x}}$ are different, we obtain much more freedom in constructing our models.

For using this model in our cubically regularized method (2), we only need to define the gradient $\boldsymbol{g} = \mathcal{G}(h,\boldsymbol{x},\tilde{\boldsymbol{x}})$ and the Hessian estimates $\boldsymbol{H} = \mathcal{H}(h,\boldsymbol{x},\tilde{\boldsymbol{x}})$, and we can also treat them differently (using two different helpers $h_1$ and $h_2$, correspondingly). Thus we come to the following general second-order (meta)algorithm. We perform $S$ rounds, the length of each round is $m \geq 1$, which is our key parameter:

---
**Algorithm 1** Cubic Newton with helper functions

---
**Input:** $\boldsymbol{x}_0 \in \mathbb{R}^d$, $S$, $m \geq 1$, $M > 0$.
1: **for** $t = 0, \ldots, Sm - 1$ **do**
2:     **if** $t \bmod m = 0$ **then**
3:         Update $\tilde{\boldsymbol{x}}_t$ (using previous states $\boldsymbol{x}_{i\leq t}$)
4:     **else**
5:         $\tilde{\boldsymbol{x}}_t = \tilde{\boldsymbol{x}}_{t-1}$
6:     Form helper functions $h_1, h_2$
7:     Compute the gradient $\boldsymbol{g}_t = \mathcal{G}(h_1, \boldsymbol{x}_t, \tilde{\boldsymbol{x}}_t)$, and the Hessian $\boldsymbol{H}_t = \mathcal{H}(h_2, \boldsymbol{x}_t, \tilde{\boldsymbol{x}}_t)$
8:     Compute the cubic step $\boldsymbol{x}_{t+1} \in \arg\min_{\boldsymbol{y}\in\mathbb{R}^d} \Omega_{M,\boldsymbol{g}_t,\boldsymbol{H}_t}(\boldsymbol{y}, \boldsymbol{x}_t)$
    **return** $\boldsymbol{x}_{out}$ using the history $(\boldsymbol{x}_i)_{0\leq i\leq Sm}$

---

In Algorithm 1 we update the snapshot $\tilde{\boldsymbol{x}}$ regularly every $m$ iterations. The two possible options are

$$\tilde{\boldsymbol{x}}_t \quad = \quad \boldsymbol{x}_{t \bmod m} \qquad\qquad \text{(use the last iterate)} \tag{4}$$

or

$$\tilde{\boldsymbol{x}}_t \quad = \quad \arg\min_{i\in\{t-m+1,\ldots,t\}} f(\boldsymbol{x}_i) \qquad \text{(use the best iterate)} \tag{5}$$

Clearly, option (5) is available only in case we can efficiently estimate the function values. However, we will see that it serves us with better global convergence guarantees, for the gradient-dominated functions.

It remains only to specify how we choose the helpers $h_1$ and $h_2$. We need to assume that they are somehow similar to $f$. Let us present several efficient choices that lead to implementable second-order schemes.

## 3.1 Basic Stochastic Methods

If the objective function $f$ is very "expensive" (for example of the form (1) with $n \to \infty$), one option is to ignore the part $f - h$ i.e. to approximate it by a zeroth-order approximation: $f(\boldsymbol{y}) - h(\boldsymbol{y}) \approx f(\tilde{\boldsymbol{x}}) - h(\tilde{\boldsymbol{x}})$. Since it is just a constant, we do not need to update $\tilde{\boldsymbol{x}}$. In this case, we have:

$$\mathcal{G}(h_1, \boldsymbol{x}, \tilde{\boldsymbol{x}}) \; := \; \nabla h_1(\boldsymbol{x}), \qquad \mathcal{H}(h_2, \boldsymbol{x}, \tilde{\boldsymbol{x}}) \; := \; \nabla^2 h_2(\boldsymbol{x}).$$

To treat this choice of the helpers and motivated by the form of the errors in Lemma 5, we assume the following similarity assumptions:

---

**Assumption 2 (Bounded similarity)** *Let for some $\delta_1, \delta_2 \geq 0$, it holds*

$$\mathbb{E}_{h_1}[\|\mathcal{G}(h_1, \boldsymbol{x}, \tilde{\boldsymbol{x}}) - \nabla f(\boldsymbol{x})\|^{3/2}] \leq \delta_1^{3/2}, \; \mathbb{E}_{h_2}[\|\mathcal{H}(h_2, \boldsymbol{x}, \tilde{\boldsymbol{x}}) - \nabla^2 f(\boldsymbol{x})\|^3] \leq \delta_2^3, \quad \forall \boldsymbol{x}, \tilde{\boldsymbol{x}} \in \mathbb{R}^d.$$

---

Under this assumption, we prove the following theorem:

---

**Theorem 1** *Under Assumptions 1 and 2, and $M \geq L$, for an output of Algorithm 1 $\boldsymbol{x}_{out}$ chosen uniformly at random from $(\boldsymbol{x}_i)_{0 \leq i \leq Sm}$, we have:*

$$\mathbb{E}[\mu_M(\boldsymbol{x}_{out})] \;\; = \;\; \mathcal{O}\Big( \tfrac{\sqrt{M} F_0}{Sm} + \tfrac{\delta_2^3}{M^{3/2}} + \delta_1^{3/2} \Big).$$

---

We see that according to this result, we can get $\mathbb{E}[\mu_M(\boldsymbol{x}_{out})] \leq \varepsilon^{3/2}$ only for $\varepsilon > \delta_1$. In other words, we can converge only to a certain *neighbourhood around a stationary point*, that is determined by the error $\delta_1$ of the stochastic gradients.

However, as we will show next, this seemingly pessimistic dependence leads to the same rate of classical subsampled Cubic Newton methods discovered in [17, 32, 33].

Let us discuss now the specific case of stochastic optimization, where $f$ has the specific form (1), with $n$ potentially being very large. In this case, it is customary to sample batches at random and assume the noise to be bounded in expectation. Precisely speaking, if we assume the standard assumption that for one index sampled uniformly at random, we have $\mathbb{E}_i \|\nabla f(\boldsymbol{x}) - \nabla f_i(\boldsymbol{x})\|^2 \leq \sigma_g^2$ and $\mathbb{E}_i \|\nabla^2 f(\boldsymbol{x}) - \nabla^2 f_i(\boldsymbol{x})\|^3 \leq \sigma_h^3$, then it is possible to show that for

$$h_1 \;\; = \;\; \tfrac{1}{b_g} \sum_{i \in \mathcal{B}_g} f_i \qquad \text{and} \qquad h_2 \;\; = \;\; \tfrac{1}{b_h} \sum_{i \in \mathcal{B}_h} f_i, \tag{6}$$

for batches $\mathcal{B}_g, \mathcal{B}_h \subseteq [n]$ sampled uniformly at random and of sizes $b_g$ and $b_h$ respectively, Assumption 2 is satisfied with [27]: $\delta_1 = \frac{\sigma_g}{\sqrt{b_g}}$ and $\delta_2 = \tilde{\mathcal{O}}(\frac{\sigma_h}{\sqrt{b_h}})$. Note that we can use the same random subsets of indices $\mathcal{B}_g, \mathcal{B}_h$ for all iterations.

**Corollary 1** *In Algorithm 1, let us choose $M = L$ and $m = 1$, with basic helpers (6). Then, according to Theorem 1, for any $\varepsilon > 0$, to reach an $(\varepsilon, L)$-approximate second-order local minimum, we need at most $S = \frac{\sqrt{L} F_0}{\varepsilon^{3/2}}$ iterations with $b_g = \left(\frac{\sigma_g}{\varepsilon}\right)^2$ and $b_h = \frac{\sigma_h^2}{\varepsilon}$. Therefore, the total arithmetic complexity of the method becomes*

$$\mathcal{O}\Big( \tfrac{\sigma_g^2}{\varepsilon^{7/2}} + \tfrac{\sigma_h^2}{\varepsilon^{5/2}} d_{\text{eff}} \Big) \times GradCost.$$

It improves upon the complexity $\mathcal{O}(\frac{1}{\varepsilon^4}) \times GradCost$ of the first-order SGD for non-convex optimization [11], unless $d_{\text{eff}} > \frac{1}{\varepsilon^{3/2}}$ (high cost of computing the Hessians).

## 3.2 Let the Objective Guide Us

If the objective $f$ is such that we can afford to access its gradients and Hessians from time to time (functions of the form (1) with $n < \infty$ and "reasonable"), then we can do better than the previous chapter. In this case, we can afford to use a better approximation of the term $f(\boldsymbol{y}) - h(\boldsymbol{y})$. From a theoretical point of view, we can treat the case when $f$ is only differentiable once, and thus we can only use a first-order approximation of $f - h$, in this case, we will only be using the hessian of the helper $h$ but only gradients of $f$. However, in our case, if we assume we have access to gradients then

we can also have access to the Hessians of $f$ as well (from time to time). For this reason, we consider a second-order approximation of the term $f - h$, if we follow the procedure that we described above we find:

$$\mathcal{G}(h_1, \boldsymbol{x}, \tilde{\boldsymbol{x}}) \quad := \quad \nabla h_1(\boldsymbol{x}) - \nabla h_1(\tilde{\boldsymbol{x}}) + \nabla f(\tilde{\boldsymbol{x}}) + (\nabla^2 f(\tilde{\boldsymbol{x}}) - \nabla^2 h_1(\tilde{\boldsymbol{x}}))(\boldsymbol{x} - \tilde{\boldsymbol{x}}) \quad (7)$$

$$\mathcal{H}(h_2, \boldsymbol{x}, \tilde{\boldsymbol{x}}) \quad := \quad \nabla^2 h_2(\boldsymbol{x}) - \nabla^2 h_2(\tilde{\boldsymbol{x}}) + \nabla^2 f(\tilde{\boldsymbol{x}}) \quad (8)$$

We see that there is an explicit dependence on the snapshot $\tilde{\boldsymbol{x}}$ and thus we need to address the question of how this snapshot point should be updated in Algorithm 1. In general, we can update it with a certain probability $p \sim \frac{1}{m}$, and we can use more advanced combinations of past iterates (like the average). However, for our purposes, we simply choose option 4 (i.e. the last iterate), thus it is only updated once every $m$ iterations.

We also need to address the question of the measure of similarity in this case. Since we are using a second-order approximation of $f - h$, it is very logical to compare them using the difference between their third derivatives or equivalently, the Hessian Lipschitz constant of their difference. Precisely we make the following similarity assumption :

---

**Assumption 3 (Lipschitz similarity)** *Let for some $\delta_1, \delta_2 \geq 0$, it holds, $\forall \boldsymbol{x}, \tilde{\boldsymbol{x}} \in \mathbb{R}^d$:*

$$\mathbb{E}_{h_1}[\|\mathcal{G}(h_1, \boldsymbol{x}, \tilde{\boldsymbol{x}}) - \nabla f(\boldsymbol{x})\|^{3/2}] \quad \leq \quad \delta_1^{3/2} \|\boldsymbol{x} - \tilde{\boldsymbol{x}}\|^3,$$

$$\mathbb{E}_{h_2}[\|\mathcal{H}(h_2, \boldsymbol{x}, \tilde{\boldsymbol{x}}) - \nabla^2 f(\boldsymbol{x})\|^3] \quad \leq \quad \delta_2^3 \|\boldsymbol{x} - \tilde{\boldsymbol{x}}\|^3.$$

---

In particular, if $f - h_1$ and $f - h_2$ have $\delta_1$ and $\delta_2$ Lipschitz Hessians respectively then $h_1$ and $h_2$ satisfy Assumption 3.

Under this assumption, we show that the errors resulting from the use of the snapshot can be successfully balanced by choosing $M$ satisfying:

$$4\left(\frac{\delta_1}{M}\right)^{3/2} + 73\left(\frac{\delta_2}{M}\right)^3 \quad \leq \quad \frac{1}{24m^3}. \quad (9)$$

And we have the following theorem.

---

**Theorem 2** *For $f, h_1, h_2$ verifying Assumptions 1,3. For a regularization parameter $M$ chosen such that $M \geq L$ and (9) is satisfied. For an output of Algorithm 1 $\boldsymbol{x}_{out}$ chosen uniformly at random from $(\boldsymbol{x}_i)_{0 \leq i \leq Sm := T}$, we have:*

$$\mathbb{E}[\mu_M(\boldsymbol{x}_{out})] \quad = \quad \mathcal{O}\left(\frac{\sqrt{M}F_0}{Sm}\right),$$

---

In particular, we can choose $M = \max(L, 32\delta_1 m^2, 16\delta_2 m)$ which gives

$$\mathbb{E}[\mu_M(\boldsymbol{x}_{out})] \quad = \quad \mathcal{O}\left(\frac{\sqrt{L}F_0}{Sm} + \frac{\sqrt{\delta_2}F_0}{S\sqrt{m}} + \frac{\sqrt{\delta_1}F_0}{S}\right). \quad (10)$$

Based on the choices of the helpers $h_1$ and $h_2$ we can have many algorithms. We discuss these in the following sections. We start by discussing variance reduction and Lazy Hessians which rely on sampling batches randomly, then move to core-sets which try to find, more intelligently, representative weighted batches of data, after this, we discuss semi-supervised learning and how unlabeled data can be used to engineer the helpers. More generally, auxiliary learning tries to leverage auxiliary tasks in training a given main task, the auxiliary tasks can be treated as helpers.

## 3.3 Variance Reduction and Lazy Hessians

The following lemma demonstrates that we can create helper functions $h$ with lower similarity to the main function $f$ of the form (1) by employing sampling and averaging.

> **Lemma 1** *Let $f = \frac{1}{n}\sum_{i=1}^{n} f_i$ such that all $f_i$ are twice differentiable and have L-Lipschitz Hessians. Let $\mathcal{B} \subset \{1, \cdots, n\}$ be of size $b$ and sampled with replacement uniformly at random, and define $h_{\mathcal{B}} = \frac{1}{b}\sum_{i\in\mathcal{B}} f_i$, then $h_{\mathcal{B}}$ satisfies Assumption 3 with $\delta_1 = \frac{L}{\sqrt{b}}$ and $\delta_2 = \mathcal{O}(\frac{\sqrt{\log(d)}L}{\sqrt{b}})$.*

**Choice of the parameter $m$ in Algorithm 1.** Minimizing the total arithmetic cost, we choose $m = \arg\min_m \#Grad(m,\varepsilon) + d\#Hess(m,\varepsilon)$, where $\#Grad(m,\varepsilon)$ and $\#Hess(m,\varepsilon)$ denote the number of gradients and Hessians required to find an $\varepsilon$ stationary point.

Now we are ready to discuss several special cases that are direct consequences from Theorem 2.

First, note that choosing $h_1 = h_2 = f$ gives the classical Cubic Newton method [20], whereas choosing $h_1 = f$ and $h_2 = 0$, gives the Lazy Cubic Newton [9]. In both cases, we recuperate the known rates of convergence.

**General variance reduction.** If we sample batches $\mathcal{B}_g$ and $\mathcal{B}_h$ of sizes $b_g$ and $b_h$ consecutively at random and choose

$$ h_1 = \frac{1}{b_g}\sum_{i\in\mathcal{B}_g} f_i \qquad \text{and} \qquad h_2 = \frac{1}{b_h}\sum_{i\in\mathcal{B}_h} f_i, $$

and use these helpers along with the estimates (7), (8), we obtain the *Variance Reduced Cubic Newton* algorithm [35, 29]. According to Lemma 1, this choice corresponds to $\delta_1 = \frac{L}{\sqrt{b_g}}$ and $\delta_2 = \tilde{\mathcal{O}}(\frac{L}{\sqrt{b_h}})$. For $b_g \sim m^4 \wedge n, b_h \sim m^2 \wedge n$ and $M = L$, we have the non-convex convergence rate $\mathcal{O}\big(\frac{\sqrt{L}F_0}{Sm}\big)$, which is the same as that of the cubic Newton algorithm but with a smaller cost per iteration. Minimizing the total arithmetic cost, we can choose $m = \arg\min_m \frac{dn+d(m^3\wedge nm)+(m^5\wedge nm)}{m}$. Let us denote by $g^{VR}(n,d)$ the corresponding optimal value. Then we reach an $(\varepsilon, L)$-approximate second-order local minimum in at most $\mathcal{O}(\frac{g^{VR}(n,d)}{\varepsilon^{3/2}}) \times GradCost$ arithmetic operations.

**Variance reduction with Lazy Hessians.** We can also use lazy updates for Hessians combined with variance-reduced gradients. This corresponds to choosing

$$ h_1 = \frac{1}{b_g}\sum_{i\in\mathcal{B}_g} f_i \qquad \text{and} \qquad h_2 = 0, $$

which implies (according to Lemma 1) that $\delta_1 = \frac{L}{\sqrt{b_g}}$ and $\delta_2 = L$. In this case, we need $b_g \sim m^2$ to obtain a convergence rate of $\mathcal{O}\big(\frac{\sqrt{L}F_0}{S\sqrt{m}}\big)$, which matches the convergence rate of the Lazy Cubic Newton method while using stochastic gradients. We choose this time $m = \arg\min_m \frac{nd+(m^3\wedge mn)}{\sqrt{m}}$, as before. Let us denote $g^{Lazy}(n,d)$ the corresponding minimum. Then we guarantee to reach an $(\varepsilon, mL)$-approximate second-order local minimum in at most $\mathcal{O}(\frac{g^{Lazy}(n,d)}{\varepsilon^{3/2}}) \times GradCost$ operations.

**To be lazy or not to be?** We show that $g^{Lazy}(n,d) \sim (nd)^{5/6} \wedge n\sqrt{d}$ and $g^{VR}(n,d) \sim (nd)^{4/5} \wedge (n^{2/3}d + n)$. In particular, for $d \geq n^{2/3}$ we have $g^{Lazy}(n,d) \leq g^{VR}(n,d)$ and thus for $d \geq n^{2/3}$ *it is better to use Lazy Hessians* than variance-reduced Hessians from a gradient equivalent cost perspective. We note also that for the Lazy approach, we can keep a factorization of the Hessian (this factorization induces most of the cost of solving the cubic subproblem) and thus it is as if we only need to solve the subproblem once every $m$ iterations, so the Lazy approach has a big advantage compared to the general approach, and the advantage becomes even bigger for the case of large dimensions.

Note that according to the theory, we could use the same random batches $\mathcal{B}_g, \mathcal{B}_h \subseteq [n]$ generated once for all iterations. However, using the resampled batches can lead to a more stable convergence.

### 3.4 Other Applications

The result in (10) is general enough that it can include many other applications that are only limited by our imagination. To cite a few such applications there are:

**Core sets.** [3] The idea of core sets is simple: can we summarize a potentially big data set using only a few (weighted) important examples? Many reasons such as redundancy make the answer yes. Devising approaches to find such core sets is outside of the scope of this work, but in general, we can see from (10) that if we have batches $\mathcal{B}_g, \mathcal{B}_h$ such that they are $(\delta_1, 1)$ and $(\delta_2, 2)$ similar to $f$ respectively, then we can keep reusing the same batch $\mathcal{B}_g$ for at least $\sqrt{\frac{L}{\delta_1}}$ times, and $\mathcal{B}_h$ for $\frac{L}{\delta_2}$ all the while guaranteeing an improved rate. So then if we can design such small batches with small $\delta_1$ and $\delta_2$ then we can keep reusing them, and joy the improved rate without needing large batches.

**Auxiliary learning.** [4, 2, 31] study how a given task $f$ can be trained in the presence of auxiliary (related) tasks. Our approach can be indeed used for auxiliary learning by treating the auxiliaries as helpers. If we compare (10) to the rate that we obtained without the use of the helpers: $\mathcal{O}(\frac{\sqrt{L}F_0}{S})$, we see that we have a better rate using the helpers/auxiliary tasks when $\frac{1}{m} + \frac{\sqrt{\delta_2}}{\sqrt{mL}} + \frac{\sqrt{\delta_1}}{\sqrt{L}} \leq 1$.

**Semi-supervised learning.**[34] Semi-supervised learning is a machine learning approach that combines the use of both labeled data and unlabeled data during training. In general, we can use the unlabeled data to construct the helpers, we can start for example by using random labels for the helpers and improving the labels with training. There are at least two special cases where our theory implies improvement by only assigning random labels to the unlabeled data. In fact, for both regularized least squares and logistic regression, we notice that the Hessian is independent of the labels (only depends on inputs) and thus if the unlabeled data comes from the same distribution as the labeled data, then we can use it to construct helpers which, at least theoretically, have $\delta_1 = \delta_2 = 0$. Because the Hessian is independent of the labels, we can technically endow the unlabeled data with random labels. Theorem 2 would imply in this case $\mathbb{E}[\mu_L(\boldsymbol{x}_{out})] = \mathcal{O}(\frac{\sqrt{L}F_0}{Sm})$, where $S$ is the number of times we use labeled data and $S(m-1)$ is the number of unlabeled data.

# 4 Gradient-Dominated Functions

We consider now the class of gradient-dominated functions defined below.

---

**Assumption 4** $(\tau, \alpha)$-**gradient dominated.** *A function $f$ is called gradient dominated on set if it holds, for some $\alpha \geq 1$ and $\tau > 0$:*

$$f(\boldsymbol{x}) - f^\star \quad \leq \quad \tau\|\nabla f(\boldsymbol{x})\|^\alpha, \qquad \forall \boldsymbol{x} \in \mathbb{R}^d. \tag{11}$$

---

Examples of functions satisfying this assumption are convex functions ($\alpha = 1$) and strongly convex functions ($\alpha = 2$), see Appendix D.1. For such functions, we can guarantee convergence (in expectation) to a *global minimum*, i.e. we can find a point $\boldsymbol{x}$ such that $f(\boldsymbol{x}) - f^\star \leq \varepsilon$.

The Gradient-dominance property is interesting because many non-convex functions have been shown to satisfy it [28, 13, 18]. Furthermore, besides convergence to a global minimum, we get accelerated rates.

We note that for $\alpha > 3/2$ (and only for this case), we needed to assume the following (stronger) inequality:

$$\mathbb{E}f(\boldsymbol{x}_t) - f^\star \quad \leq \quad \tau\mathbb{E}\big[\|\nabla f(\boldsymbol{x}_t)\|\big]^\alpha, \tag{12}$$

where the expectation is taken with respect to the iterates $(\boldsymbol{x}_t)$ of our algorithms. This is a stronger assumption than (11). To avoid using this stronger assumption, we can assume that the iterates belong to some compact set $Q \subset \mathbb{R}^d$ and that the gradient norm is uniformly bounded: $\forall \boldsymbol{x} \in Q : \|\nabla f(\boldsymbol{x})\| \leq G$. Then, a $(\tau, \alpha)$-gradient dominated on set $Q$ function is also a $(\tau G^{\alpha-3/2}, 3/2)$-gradient dominated on this set for any $\alpha > 3/2$.

In Theorem 3 we extend the results of Theorem 1 to gradient-dominated functions.

> **Theorem 3** *Under Assumptions 1,2,4, for $M \geq L$ and $T := Sm$ we have:*
> *- For $1 \leq \alpha \leq 3/2$:* $\mathbb{E}[f(\boldsymbol{x}_T)] - f^\star = \mathcal{O}\Big( \big( \frac{\alpha\sqrt{M}\tau^{3/(2\alpha)}}{(3-2\alpha)T} \big)^{\frac{2\alpha}{3-2\alpha}} + \tau\frac{\delta_2^{2\alpha}}{M^\alpha} + \tau\delta_1^\alpha \Big).$
>
> *- For $3/2 < \alpha \leq 2$, let $h_0 = \mathcal{O}(\frac{F_0}{(\sqrt{M}\tau^{\frac{3}{2\alpha}})^{\frac{2\alpha}{3-2\alpha}}})$, then for $T \geq t_0 = \mathcal{O}(h_0^{\frac{3-2\alpha}{2\alpha}}\log(h_0))$ we have:*
>
> $$ E[f(\boldsymbol{x}_T)] - f^\star = \mathcal{O}\Big( (\sqrt{M}\tau^{\frac{3}{2\alpha}})^{\frac{2\alpha}{3-2\alpha}} \big(\tfrac{1}{2}\big)^{(\frac{2\alpha}{3})^{T-t_0}} + \tau\frac{\delta_2^{2\alpha}}{M^\alpha} + \tau\delta_1^\alpha \Big). $$

Theorem 3 shows (up to the noise level) for $1 \leq \alpha < 3/2$ a sublinear rate, for $\alpha = 3/2$ a linear rate (obtained by taking the limit $\alpha \to 3/2$) and a superlinear rate for $\alpha > 3/2$.

We do the same thing for Theorem 2 which we extend in Theorem 4. In this case, we need to set the snapshot line 3 in Algorithm 1) as in 5 i.e. the snapshot corresponds to the state with the smallest value of $f$ during the last $m$ iterations.

> **Theorem 4** *Under Assumptions 1,3,4, for $M = \max(L, 34\delta_1 m^2, 11\delta_2 m)$, we have:*
> *- For $1 \leq \alpha \leq 3/2$ :* $\mathbb{E}[f(\boldsymbol{x}_{Sm})] - f^\star = \mathcal{O}\Big( \big( \frac{\alpha\sqrt{M}\tau^{3/(2\alpha)}}{(3-2\alpha)Sm} \big)^{\frac{2\alpha}{3-2\alpha}} \Big).$
>
> *- For $3/2 < \alpha \leq 2$, let $h_0 = \mathcal{O}(\frac{F_0}{(\frac{\sqrt{M}}{m}\tau^{\frac{3}{2\alpha}})^{\frac{2\alpha}{3-2\alpha}}})$, then for $S \geq s_0 = \mathcal{O}(h_0^{\frac{3-2\alpha}{2\alpha}}\log(h_0))$ we have:*
> $$ \mathbb{E}[f(\boldsymbol{x}_{Sm})] - f^\star = \Big( (\tfrac{\sqrt{M}}{m}\tau^{\frac{3}{2\alpha}})^{\frac{2\alpha}{3-2\alpha}} \big(\tfrac{1}{2}\big)^{(\frac{2\alpha}{3})^{S-s_0}} \Big) $$

Again, the same behavior is observed as for Theorem 3 but this time without noise (variance reduction is working). To the best of our knowledge, this is the first time such analysis is made. As a direct consequence of our results, we obtain new global complexities for the variance-reduced and lazy variance-reduced Cubic Newton methods on the class of gradient-dominated functions.

To compare the statements of Theorems 3 and 4, for convex functions (i.e. $\alpha = 1$), Theorem 3 guarantees convergence to a $\varepsilon-$global minimum in at most $\mathcal{O}(\frac{1}{\varepsilon^{5/2}} + \frac{d}{\varepsilon^{3/2}})$ *GradCost*, whereas Theorem 4 only needs $\mathcal{O}(\frac{g(n,d)}{\sqrt{\varepsilon}})$ *GradCost*, where $g(n,d)$ is either $g^{Lazy}(n,d) = (nd)^{5/6} \wedge n\sqrt{d}$ or $g^{VR}(n,d) = (nd)^{4/5} \wedge (n^{2/3}d + n)$. See the Appendix D.3 for more details.

# 5 Limitations and possible extensions

**Estimating similarity between the helpers and the main function.** While we show in this work that we can have an improvement over training alone, this supposes that we know the similarity constants $\delta_1, \delta_2$, hence it will be interesting to have approaches that can adapt to such constants.

**Engineering helper functions.** Building helper task with small similarities is also an interesting idea. Besides the examples in supervised learning and core-sets that we provide, it is not evident how to do it in a generalized way.

**Using the helper to regularize the cubic subproblem.** We note that while we proposed to approximate the "cheap" part as well in Section 3, one other theoretically viable approach is to keep it intact and approximately solve a "proximal type" problem involving $h$, this will lead to replacing $L$ by $\delta$, but the subproblem is even more difficult to solve. However our theory suggests that we don't need to solve this subproblem exactly, we only need $m \geq \frac{L}{\delta}$. We do not treat this case here.

# 6 Conclusion

In this work, we proposed a general theory for using auxiliary information in the context of the cubically regularized Newton's method. Our theory encapsulates the classical stochastic methods as well as variance reduction and Lazy methods. For auxiliary learning, we showed a provable benefit compared to training alone. Besides studying the convergence for general non-convex functions for which we show convergence to approximate local minima, we also study gradient-dominated functions, for which convergence is accelerated and is to approximate global minima.

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
