$$x^+ \in \underset{y \in \mathbb{R}^d}{\arg\min} \Big\{ \hat{f}_x(y) + Mr_x(y) \Big\},$$

where $\hat{f}_x(\cdot)$ is an approximation of $f$ around current point $x$, and $r_x(y)$ is a regularizer that encodes how accurate the approximation is, and $M > 0$ is a regularization parameter. In this work, we are interested in cubically regularized second-order models of the form (2) and we use $r_x(y) := \frac{1}{6}\|y - x\|^3$.

Now let us look at how we can use a helper $h$ to construct the approximation $\hat{f}$. We notice that we can write

$$f(y) \quad := \quad \underbrace{h(y)}_{\text{cheap}} + \underbrace{f(y) - h(y)}_{\text{expensive}}$$

We discuss the actual practical choices of the helper function $h$ below. We assume now that we can afford the second-order approximation for the cheap part $h$ around the current point $x$. However, approximating the part $f - h$ can be expensive (as for example when the number of elements $n$ in finite sum (1) is huge), or even impossible (due to lack of data). Thus, we would prefer to approximate the expensive part less frequently. For this reason, let us introduce an extra *snapshot point* $\hat{x}$ that is updated less often than $x$. Then, we use it to approximate $f - h$. Another question that we still need to ask is *what order should we use for the approximation of $f - h$?* We will see that order $0$ (approximating by a constant) leads as to the basic stochastic methods, while for orders $1$ and $2$ we equip our methods with the variance reduction.

Combining the two approximations for $h$ and $f - h$ we get the following model of our objective $f$:

$$\hat{f}_{x,\tilde{x}}(y) \quad = \quad C(x, \tilde{x}) + \langle \mathcal{G}(h, x, \tilde{x}), y - x \rangle + \tfrac{1}{2} \langle \mathcal{H}(h, x, \tilde{x})(y - x), y - x \rangle, \tag{3}$$

where $C(x, \tilde{x})$ is a constant, $\mathcal{G}(h, x, \tilde{x})$ is a linear term, and $\mathcal{H}(h, x, \tilde{x})$ is a matrix. Note that if $\tilde{x} \equiv x$, then the best second-order model of the form (3) is the Taylor polynomial of degree two for $f$ around $x$, and that would give us the exact Newton-type method. However, when the points $x$ and $\tilde{x}$ are different, we obtain much more freedom in constructing our models.

For using this model in our cubically regularized method (2), we only need to define the gradient $g = \mathcal{G}(h, x, \tilde{x})$ and the Hessian estimates $H = \mathcal{H}(h, x, \tilde{x})$, and we can also treat them differently (using two different helpers $h_1$ and $h_2$, correspondingly). Thus we come to the following general second-order (meta)algorithm. We perform $S$ rounds, the length of each round is $m \geq 1$, which is our key parameter:

---

**Algorithm 1** Cubic Newton with helper functions

---

**Input:** $x_0 \in \mathbb{R}^d$, $S$, $m \geq 1$, $M > 0$.
 1: **for** $t = 0, \ldots, Sm - 1$ **do**
 2:     **if** $t \bmod m = 0$ **then**
 3:         Update $\tilde{x}_t$ (using previous states $x_{i \leq t}$)
 4:     **else**
 5:         $\tilde{x}_t = \tilde{x}_{t-1}$
 6:     Form helper functions $h_1, h_2$
 7:     Compute the gradient $g_t = \mathcal{G}(h_1, x_t, \tilde{x}_t)$, and the Hessian $H_t = \mathcal{H}(h_2, x_t, \tilde{x}_t)$
 8:     Compute the cubic step $x_{t+1} \in \arg\min_{y \in \mathbb{R}^d} \Omega_{M, g_t, H_t}(y, x_t)$

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

# 0 Experiments

## 0.1 To be lazy or not

To verify our findings from Subsection 3.3, we consider a logistic regression problem on the "a9a" data set [7]. Specifically

$$f(\boldsymbol{x}) = \frac{1}{N} \sum_{i=1}^{N} \log(1 + \exp(-b_i \boldsymbol{x}^\top \boldsymbol{a}_i)) + \frac{\lambda}{2} \|\boldsymbol{x}\|^2 \ ,$$

where $\{(\boldsymbol{a}_i, b_i)\}_{i=1}^{N}$ are samples from our data, and $\lambda \geq 0$ is a regularization parameter.

We consider the variance-reduced cubic Newton method from [36] (referred to as "full VR"), its lazy version where we do not update the snapshot Hessian ("Lazy VR"), the stochastic Cubic Newton method ("SCN"), the Cubic Newton algorithm ("CN"), Gradient Descent with line search ("GD") and Stochastic Gradient Descent ("SGD"). We report the results in terms of time and gradient arithmetic computations needed to arrive at a given level of convergence.

Figure 1 shows how the lazy version saves both time and arithmetic computations without sacrificing the convergence precision.

### Logistic regression: a9a, d = 123, n = 32561, L2-regularization

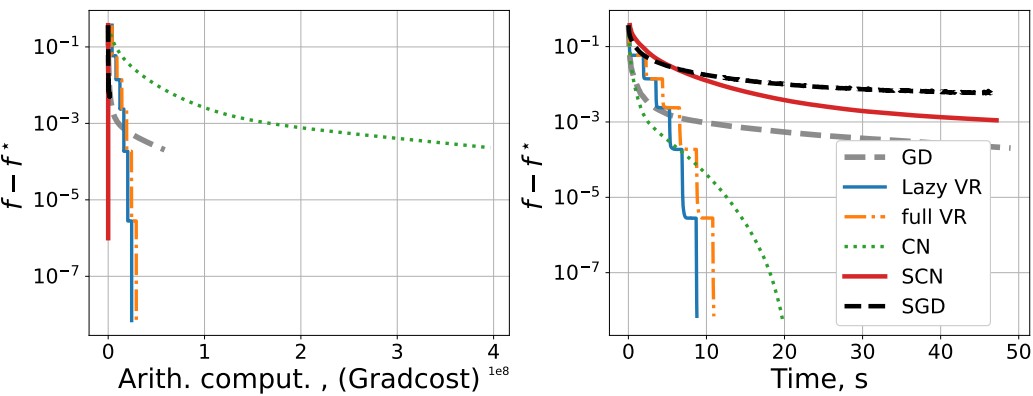

Figure 1: Comparison of the convergence of different algorithms. We see that "Lazy VR" has the same convergence speed as its full version "full VR" and the cubic Newton method "CN" while needing less time and fewer arithmetic computations.

The *Gradcost* is computed using the convention that computing one hessian is $d$ times as expensive as computing one gradient.

## 0.2 Auxiliary Learning

Our goal is to show that the helper framework is very general and that it goes beyond the variance reduction and lazy Hessian computations. For the previously considered problem of training the logistic regression (using the same "a9a" data set), we suppose that we also have access to unlabeled data (in this sense this becomes semi-supervised learning). Specifically, we have a labeled dataset $\mathcal{D}_l = \{(\boldsymbol{a}_i, b_i)\}_{i=1}^{N_l}$ and an unlabeled data set $\mathcal{D}_u = \{\boldsymbol{a}_i\}_{i=N_l+1}^{N_l+N_u}$, we suppose that both data sets are sampled from the same distribution $\mathcal{P}_{(\boldsymbol{a},\boldsymbol{b})}$.

Our goal is to minimize

$$f(\boldsymbol{x}) = \mathbb{E}_{(\boldsymbol{a},\boldsymbol{b}) \sim \mathcal{P}_{(\boldsymbol{a},\boldsymbol{b})}}[\log(1 + \exp(-b\boldsymbol{x}^\top \boldsymbol{a}))] \ .$$

A simple computation shows that the Hessian of $f$ only depends on $\mathcal{P}_{\boldsymbol{a}}$, and, for this reason, we can use unlabeled data to construct a good approximation of the true Hessian (if we can sample from $\mathcal{P}_{\boldsymbol{a}}$, we construct the exact Hessian and thus have a helper $h$ with $\delta_1 = \delta_2 = 0$).

Figure 2: Cubic Newton method with and without using the helper function $h$. For $m = 1$ this is simply the classic Cubic Newton method. To give an intuitive meaning to the plot, $\frac{1}{m}$ is the percentage of labeled data used during training. We can clearly see that using our approach we benefit a lot from the helper function $h$.

Let

$$h(\boldsymbol{x}) = \mathbb{E}_{\boldsymbol{a} \sim \mathcal{P}_{\boldsymbol{a}}, \boldsymbol{b} \sim Random\{\pm 1\}} \left[ \log(1 + \exp(-b\boldsymbol{x}^\top \boldsymbol{a})) \right],$$

where $Random\{\pm 1\}$ is any distribution on labels. In our experiments, we use uniform distribution.

Figure 2 shows that, indeed we can benefit a lot from using this helper function.

## 0.3 Additional experiments

We go back to comparing the algorithms in 0.1. We consider now non-convex problems.

First we consider logistic regression with a non-convex regularizer $Reg(\boldsymbol{x}) = \sum_{i=1}^d \frac{\boldsymbol{x}_i^2}{1 + \boldsymbol{x}_i^2}$. Precisely speaking we minimize

$$f(\boldsymbol{x}) = \frac{1}{N} \sum_{i=1}^N \log(1 + \exp(-b_i \boldsymbol{x}^\top \boldsymbol{a}_i)) + \lambda Reg(\boldsymbol{x}).$$

Figure 3 shows the results in this case. Again we see that "lazy VR" reduces both time and gradient equivalent computations without sacrificing the convergence speed.

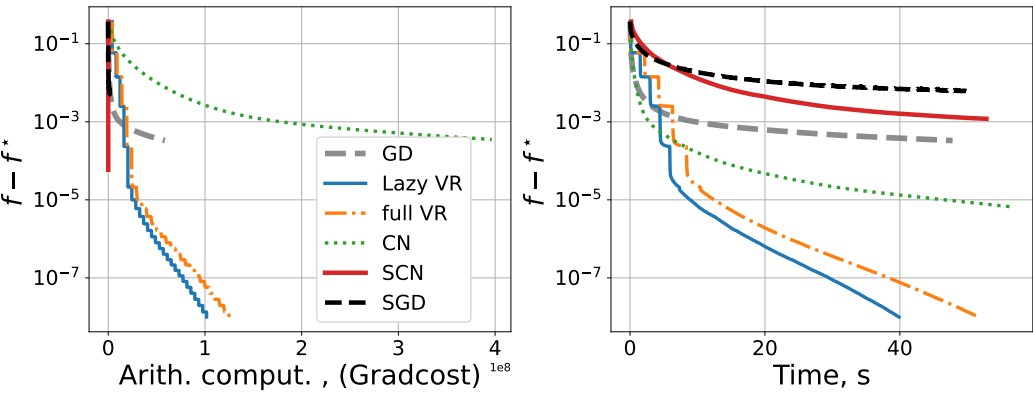

Figure 3: Comparison of the convergence of different algorithms. We see that using our approach we benefit a lot from the helper function $h$.

Second, we consider a simple diagonal neural network with L2 loss with data generated from a normal distribution. specifically, we want to minimize

$$f(\boldsymbol{x} := (\boldsymbol{u}, \boldsymbol{v})) = \frac{1}{N} \sum_{i=1}^{N} \|\boldsymbol{a}_i^\top \boldsymbol{u} \circ \boldsymbol{v} - b_i\|^2 + \frac{\lambda}{2} \|\boldsymbol{x}\|^2 ,$$

where $\circ$ is the element-wise vector product.

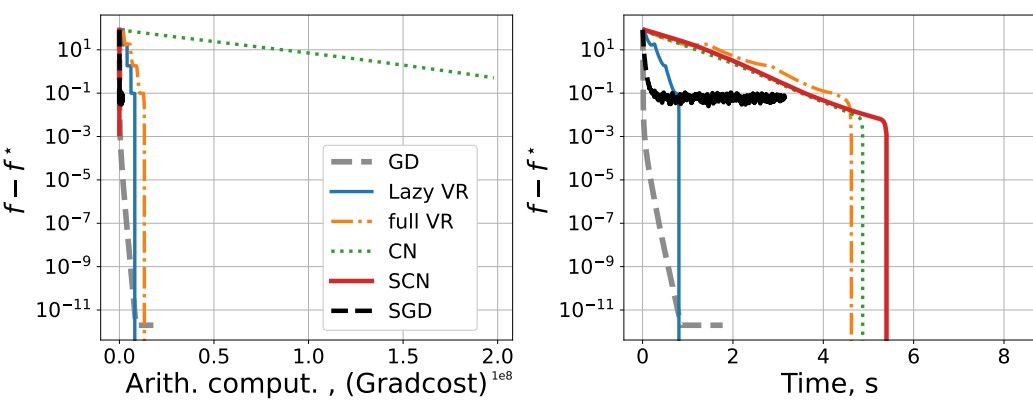

Figure 4: Comparison of the convergence of the different algorithms. Except for gradient descent ("GD") which performs very well in this case, again the same conclusions as in Figure 2 with respect to "Lazy VR" can be said.

Figure 4 shows again that compared to other second-order methods, "Lazy VR" has considerable time and computation savings. It also has a close performance to gradient descent with line search which performs very well in this case.

## 0.4 Reproducibility

We will make our code available with all the details necessary for reproducing our results in https://anonymous.4open.science/r/Unified-Convergence-Theory-of-Cubic-Newton-s-method-E4C0/README.md.

## A    Theoretical Preliminaries

We consider the general problem

$$\min_{\boldsymbol{x} \in \mathbb{R}^d} f(\boldsymbol{x})$$

Where $f$ is twice differentiable with $L$-Lipschitz Hessian i.e.:

$$\|\nabla^2 f(\boldsymbol{x}) - \nabla^2 f(\boldsymbol{y})\| \leq L\|\boldsymbol{x} - \boldsymbol{y}\|, \qquad \forall \boldsymbol{x}, \boldsymbol{y} \in \mathbb{R}^d. \tag{13}$$

As a direct consequence of (13) (see [21, 20]) we have for all $\boldsymbol{x}, \boldsymbol{y} \in \mathbb{R}^d$:

$$\|\nabla f(\boldsymbol{y}) - \nabla f(\boldsymbol{x}) - \nabla^2 f(\boldsymbol{x})(\boldsymbol{y} - \boldsymbol{x})\| \leq \frac{L}{2}\|\boldsymbol{x} - \boldsymbol{y}\|^2, \tag{14}$$

$$\left| f(\boldsymbol{y}) - f(\boldsymbol{x}) - \langle \nabla f(\boldsymbol{x}), \boldsymbol{y} - \boldsymbol{x} \rangle - \frac{1}{2} \langle \nabla^2 f(\boldsymbol{x})(\boldsymbol{y} - \boldsymbol{x}), \boldsymbol{y} - \boldsymbol{x} \rangle \right| \leq \frac{L}{6}\|\boldsymbol{y} - \boldsymbol{x}\|^3. \tag{15}$$

For $\boldsymbol{x}$ and $\boldsymbol{x}^+$ defined as in Equation (2) i.e.

$$\boldsymbol{x}^+ \in \arg\min_{\boldsymbol{y} \in \mathbb{R}^d} \left\{ \Omega_{M,\boldsymbol{g},\boldsymbol{H}}(\boldsymbol{y}, \boldsymbol{x}) := \langle \boldsymbol{g}, \boldsymbol{y} - \boldsymbol{x} \rangle + \tfrac{1}{2} \langle \boldsymbol{H}(\boldsymbol{y} - \boldsymbol{x}), \boldsymbol{y} - \boldsymbol{x} \rangle + \tfrac{M}{6}\|\boldsymbol{y} - \boldsymbol{x}\|^3 \right\}. \tag{16}$$

The optimality condition of (16) ensures that :

$$\langle \boldsymbol{g}, \boldsymbol{x}^+ - \boldsymbol{x} \rangle + \langle \boldsymbol{H}(\boldsymbol{x}^+ - \boldsymbol{x}), \boldsymbol{x}^+ - \boldsymbol{x} \rangle + \frac{M}{2}r^3 = 0, \tag{17}$$

where we denoted $r = \|\boldsymbol{x}^+ - \boldsymbol{x}\|$.

It is also known that the solution to (16) verifies :

$$\boldsymbol{H} + \frac{M}{2}r\mathbb{I} \succeq 0 \tag{18}$$

We start by proving the following Theorem

> **Theorem 5** *For any $\boldsymbol{x} \in \mathbb{R}^d$, let $\boldsymbol{x}^+$ be defined by (2). Then, for $M \geq L$ we have:*
>
> $$f(\boldsymbol{x}) - f(\boldsymbol{x}^+) \geq \frac{1}{1008\sqrt{M}}\mu_M(\boldsymbol{x}^+) + \frac{M\|\boldsymbol{x} - \boldsymbol{x}^+\|^3}{72} - \frac{4\|\nabla f(\boldsymbol{x}) - \boldsymbol{g}\|^{3/2}}{\sqrt{M}} - \frac{73\|\nabla^2 f(\boldsymbol{x}) - \boldsymbol{H}\|^3}{M^2}.$$

Using (15) with $\boldsymbol{y} = \boldsymbol{x}^+$ and $\boldsymbol{x} = \boldsymbol{x}$ and for $M \geq L$ we have:

$$
\begin{aligned}
f(\boldsymbol{x}^+) &\overset{(15)}{\leq} f(\boldsymbol{x}) + \langle \nabla f(\boldsymbol{x}), \boldsymbol{x}^+ - \boldsymbol{x} \rangle + \tfrac{1}{2}\langle \nabla^2 f(\boldsymbol{x})(\boldsymbol{x}^+ - \boldsymbol{x}), \boldsymbol{x}^+ - \boldsymbol{x} \rangle + \tfrac{L}{6}r^3 \\
&\overset{(17)+(18)}{\leq} f(\boldsymbol{x}) - \tfrac{6M - 4L}{24}r^3 + \langle \nabla f(\boldsymbol{x}) - \boldsymbol{g}, \boldsymbol{x}^+ - \boldsymbol{x} \rangle + \tfrac{1}{2}\langle(\nabla^2 f(\boldsymbol{x}) - \boldsymbol{H})(\boldsymbol{x}^+ - \boldsymbol{x}), \boldsymbol{x}^+ - \boldsymbol{x} \rangle \\
&\overset{M \geq L}{\leq} f(\boldsymbol{x}) - \tfrac{M}{12}r^3 + \langle \nabla f(\boldsymbol{x}) - \boldsymbol{g}, \boldsymbol{x}^+ - \boldsymbol{x} \rangle + \tfrac{1}{2}\langle(\nabla^2 f(\boldsymbol{x}) - \boldsymbol{H})(\boldsymbol{x}^+ - \boldsymbol{x}), \boldsymbol{x}^+ - \boldsymbol{x} \rangle
\end{aligned}
$$

Using Young's inequality $xy \leq \frac{x^p}{p} + \frac{y^q}{q} \ \forall x, y \in \mathbb{R} \ \forall p, q > 0$ s.t $\frac{1}{p} + \frac{1}{q} = 1$ we have:

$$\langle \nabla f(\boldsymbol{x}) - \boldsymbol{g}, \boldsymbol{x}^+ - \boldsymbol{x} \rangle \leq \frac{M}{36}r^3 + \frac{2\sqrt{12}}{3\sqrt{M}}\|\nabla f(\boldsymbol{x}) - \boldsymbol{g}\|^{3/2},$$

and

$$\frac{1}{2}\langle(\nabla^2 f(\boldsymbol{x}) - \boldsymbol{H})(\boldsymbol{x}^+ - \boldsymbol{x}), \boldsymbol{x}^+ - \boldsymbol{x} \rangle \leq \frac{M}{36}r^3 + \frac{72}{M^2}\|\nabla^2 f(\boldsymbol{x}) - \boldsymbol{H}\|^3.$$

Mixing all these ingredients, we get

> **Lemma 2** *For any $M \geq L$, it holds*
>
> $$f(\boldsymbol{x}) - f(\boldsymbol{x}^+) \geq \frac{M}{36}r^3 - \frac{3}{\sqrt{M}}\|\nabla f(\boldsymbol{x}) - \boldsymbol{g}\|^{3/2} - \frac{72}{M^2}\|\nabla^2 f(\boldsymbol{x}) - \boldsymbol{H}\|^3. \tag{19}$$

Using (14) we have:

$$\|\nabla f(\boldsymbol{x}^+) - \boldsymbol{g} - \boldsymbol{H}(\boldsymbol{x}^+ - \boldsymbol{x}) + \boldsymbol{g} - \nabla f(\boldsymbol{x}) + (\boldsymbol{H} - \nabla^2 f(\boldsymbol{x}))(\boldsymbol{x}^+ - \boldsymbol{x})\| \leq \frac{L}{2} r^2,$$

applying the triangular inequality we get for $M \geq L$ :

$$\begin{aligned}
\|\nabla f(\boldsymbol{x}^+)\| &\leq \frac{L}{2} r^2 + \|\boldsymbol{g} + \boldsymbol{H}(\boldsymbol{x}^+ - \boldsymbol{x})\| + \|\nabla f(\boldsymbol{x}) - \boldsymbol{g}\| + \|\nabla^2 f(\boldsymbol{x}) - \boldsymbol{H}\| r \\
&\leq \frac{L + 2M}{2} r^2 + \|\nabla f(\boldsymbol{x}) - \boldsymbol{g}\| + \frac{1}{2M} \|\nabla^2 f(\boldsymbol{x}) - \boldsymbol{H}\|^2 \\
&\leq \frac{3M}{2} r^2 + \|\nabla f(\boldsymbol{x}) - \boldsymbol{g}\| + \frac{1}{2M} \|\nabla^2 f(\boldsymbol{x}) - \boldsymbol{H}\|^2
\end{aligned}$$

By the convexity of $x \mapsto x^{3/2}$ we have for any $(a_i) \geq 0$ : $(\sum_i a_i x_i)^{3/2} \leq (\sum_i a_i)^{1/2} \sum_i a_i x_i^{3/2}$, applying this to the above inequality we get

**Lemma 3** *For any $M \geq L$, it holds*

$$\frac{1}{\sqrt{M}} \|\nabla f(\boldsymbol{x}^+)\|^{3/2} \leq 3M r^3 + \frac{2}{\sqrt{M}} \|\nabla f(\boldsymbol{x}) - \boldsymbol{g}\|^{3/2} + \frac{1}{M^2} \|\nabla^2 f(\boldsymbol{x}) - \boldsymbol{H}\|^3 \qquad (20)$$

We can also bound the smallest eigenvalue of the Hessian. Using the smoothness of the Hessian we have:

$$\begin{aligned}
\nabla^2 f(\boldsymbol{x}^+) &\succeq \nabla^2 f(\boldsymbol{x}) - L\|\boldsymbol{x}^+ - \boldsymbol{x}\|\mathbb{I} \\
&\succeq \boldsymbol{H} + \nabla^2 f(\boldsymbol{x}) - \boldsymbol{H} - Lr\mathbb{I} \\
&\succeq \boldsymbol{H} - \|\nabla^2 f(\boldsymbol{x}) - \boldsymbol{H}\|\mathbb{I} - Lr\mathbb{I} \\
&\overset{(18)}{\succeq} -\frac{Mr}{2}\mathbb{I} - \|\nabla^2 f(\boldsymbol{x}) - \boldsymbol{H}\|\mathbb{I} - Lr\mathbb{I}
\end{aligned}$$

Which means for $M \geq L$ we have:

$$-\lambda_{min}(\nabla^2 f(\boldsymbol{x}^+)) \leq \frac{3Mr}{2} + \|\nabla^2 f(\boldsymbol{x}) - \boldsymbol{H}\|$$

Then the convexity of $x \mapsto x^3$ leads to the following lemma :

**Lemma 4** *For any $M \geq L$, it holds*

$$\frac{-\lambda_{min}(\nabla^2 f(\boldsymbol{x}^+))^3}{M^2} \leq 14 M r^3 + \frac{4}{M^2} \|\nabla^2 f(\boldsymbol{x}) - \boldsymbol{H}\|^3 \qquad (21)$$

Now the quantity $\mu_M(\boldsymbol{x}) = \max(\|\nabla f(\boldsymbol{x})\|^{3/2}, \frac{-\lambda_{min}(\nabla^2 f(\boldsymbol{x}^+))^3}{M^{3/2}})$ which we can be bounded using Lemmas 3 and 4 :

$$\frac{1}{\sqrt{M}} \mu(\boldsymbol{x}^+) \leq 14 M r^3 + \frac{2}{\sqrt{M}} \|\nabla f(\boldsymbol{x}) - \boldsymbol{g}\|^{3/2} + \frac{4}{M^2} \|\nabla^2 f(\boldsymbol{x}) - \boldsymbol{H}\|^3. \qquad (22)$$

Combining Lemma 2 and (22) we get the inequality given in Theorem 5:

$$f(\boldsymbol{x}) - f(\boldsymbol{x}^+) \geq \frac{1}{1008\sqrt{M}} \mu_M(\boldsymbol{x}^+) + \frac{M}{72} r^3 - \frac{4}{\sqrt{M}} \|\nabla f(\boldsymbol{x}) - \boldsymbol{g}\|^{3/2} - \frac{73}{M^2} \|\nabla^2 f(\boldsymbol{x}) - \boldsymbol{H}\|^3.$$

# B More on Section 3.1

## B.1 Similarity using sampling

One common approach for constructing gradient and Hessian estimates is sub-sampling. The idea behind sub-sampling is simple: for an objective of the form in (1), we randomly sample two batches $\mathcal{B}_g$ and $\mathcal{B}_h$ of sizes $b_g$ and $b_h$ consecutively from the distribution $\mathcal{D}$ and define:

$$\boldsymbol{g}_{t,\mathcal{B}_g} = \frac{1}{b_g} \sum_{i \in \mathcal{B}_g} \nabla f_i(\boldsymbol{x}_t) \quad \text{and} \quad \boldsymbol{H}_{t,\mathcal{B}_h} = \frac{1}{b_h} \sum_{i \in \mathcal{B}_h} \nabla^2 f_i(\boldsymbol{x}_t) \qquad (23)$$

In this particular scenario, the "elementary" estimates $\nabla f(\boldsymbol{x}_t, \zeta)$ and $\nabla^2 f(\boldsymbol{x}_t, \zeta)$ are unbiased, and we will assume that they satisfy $\mathbb{E}_i \|\nabla f(\boldsymbol{x}) - \nabla f_i(\boldsymbol{x})\|^2 \leq \sigma_g^2$ and $\mathbb{E}_i \|\nabla^2 f(\boldsymbol{x}) - \nabla^2 f_i(\boldsymbol{x})\|^3 \leq \sigma_h^3$ ,.

**Lemma 5** *For the estimators defined in* (23) *we have:*

$$\mathbb{E}\|\nabla f(\boldsymbol{x}_t) - \boldsymbol{g}_{t,\mathcal{B}_g}\|^2 \leq \frac{\sigma_g^2}{b_g} \quad and \quad \mathbb{E}\|\nabla^2 f(\boldsymbol{x}_t) - \boldsymbol{H}_{t,\mathcal{B}_h}\|^3 \leq \mathcal{O}\big(\log(d)^{3/2}\frac{\sigma_h^3}{b_h^{3/2}}\big),$$

*where $\mathcal{O}$ hides constant multiplicative factors.*

Lemma 5 demonstrates how the utilization of batching can decrease the noise. To simplify things, we can keep in mind this straightforward rule:

If we employ a batch of size $b_a$, then we need to modify $\sigma_a$ by $\frac{\sigma_a}{\sqrt{b_a}}$ for $a \in \{g, h\}$.

Lemma 5 is based on the following two Lemmas :

**Lemma 6** *(Lyapunov's inequality) For any random variable $X$ and any $0 < s < t$ we have*

$$\mathbb{E}[|X|^s]^{1/s} \leq \mathbb{E}[|X|^t]^{1/t}.$$

and

**Lemma 7** *Suppose that $q \geq 2$, $p \geq 2$, and fix $r \geq \max(q, 2\log(p))$. Consider i.i.d. random self-adjoint matrices $Y_1, \cdots, Y_N$ with dimension $p \times p$, $\mathbb{E}[Y_i] = 0$. It holds that:*

$$\big[\mathbb{E}[\|\sum_{i=1}^N Y_i\|_2^q]\big]^{1/q} \leq 2\sqrt{er}\|\big(\sum_{i=1}^N \mathbb{E}[Y_i^2]\big)^{1/2}\|_2 + 4er\mathbb{E}[\max_i \|Y_i\|_2^q]^{1/q}.$$

Lemma 7 is taken from [36].

Now if we have $X_1, \cdots, X_b \in \mathbb{R}^d$, $b$ i.i.d vector-valued random variables such that $\mathbb{E}[X_i] = \mu$ and $\mathbb{E}[\|X_i - \mu\|^2] \leq \sigma^2$ then by applying Lemma 6 we get :

$$\mathbb{E}[\|\frac{1}{b}\sum_i X_i - \mu\|^{3/2}] \leq \mathbb{E}[\|\frac{1}{b}\sum_i X_i - \mu\|^2]^{3/4} \leq \frac{\sigma^{3/2}}{b^{3/4}}.$$

When we have $b$ i.i.d matrix-valued random variables $Y_1, \cdots, Y_b \in \mathbb{R}^{d \times d}$ such that $\mathbb{E}[Y_i] = \mu$, $\mathbb{E}[\|Y_i - \mu\|^2] \leq \sigma_2^2$ and $\mathbb{E}[\|Y_i - \mu\|^3] \leq \sigma_3^3$ (by Jensen's inequality $\sigma_2 \leq \sigma_3$), then by applying Lemma 7 we get:

$$\begin{aligned}
\mathbb{E}[\|\tfrac{1}{b}\sum_i Y_i - \mu\|^3] &\leq \left(2\sqrt{\tfrac{2e\log(d)}{b}}\sigma_2 + \tfrac{8e\log(d)}{b}\sigma_3\right)^3 \\
&= \mathcal{O}(\tfrac{\sigma_3^3}{b^{3/2}})
\end{aligned}$$

These last two inequalities are identical to the statement of Lemma 5.

## B.2 Proof of Theorem 1

We use here $\delta_1 = \sigma_g$ and $\delta_2 = \sigma_h$.

Combining both Theorem 5, Assumption 2 and Lemma 6 we get:

$$\begin{aligned}
\mathbb{E}f(\boldsymbol{x}_t) - \mathbb{E}f(\boldsymbol{x}_{t+1}) \quad &\geq \quad \tfrac{1}{1008\sqrt{M}}\mathbb{E}\mu_M(\boldsymbol{x}_{t+1}) - \tfrac{4}{\sqrt{M}}\mathbb{E}\|\nabla f(\boldsymbol{x}_t) - \boldsymbol{g}_t\|^{3/2} - \tfrac{73}{M^2}\mathbb{E}\|\nabla^2 f(\boldsymbol{x}_t) - \boldsymbol{H}_t\|^3 \\
&\underset{Lemma\ 6}{\geq} \quad \tfrac{1}{1008\sqrt{M}}\mathbb{E}\mu_M(\boldsymbol{x}_{t+1}) - \tfrac{4}{\sqrt{M}}\sigma_g^{3/2} - \tfrac{73}{M^2}\sigma_h^3
\end{aligned}$$

By summing the above inequality from $t = 0$ to $t = T - 1$ and rearranging we get:

$$\frac{1}{1008T}\sum_{t=1}^T \mathbb{E}\mu_M(\boldsymbol{x}_t) \leq \sqrt{M}\frac{\mathbb{E}f(\boldsymbol{x}_0) - \mathbb{E}f(\boldsymbol{x}_T)}{T} + 4\sigma_g^{3/2} + \frac{73}{M^{3/2}}\sigma_h^3$$

All is left is to use the fact that $\mathbb{E}f(\boldsymbol{x}_0) - \mathbb{E}f(\boldsymbol{x}_T) \leq \mathbb{E}f(\boldsymbol{x}_0) - f^\star = F_0$, and by definition of $\boldsymbol{x}_{out}$ : $\mathbb{E}\mu_M(\boldsymbol{x}_{out}) = \frac{1}{T}\sum_{t=1}^T \mathbb{E}\mu_M(\boldsymbol{x}_t)$, thus :

$$\frac{1}{1008}\mathbb{E}\mu_M(\boldsymbol{x}_{out}) \leq \frac{\sqrt{M}F_0}{T} + \frac{73}{M^{3/2}}\sigma_h^3 + 4\sigma_g^{3/2}$$

# C   Helper

## C.1   Proof of Lemma 1

We have

$$f = \frac{1}{n}\sum_{i=1}^{n}f_i$$

and we suppose that all the $f_i$'s have $L-$Lipschitz Hessians, so $f$ too has an $L-$Lipschitz Hessian.

Thus $f_i - f$ has $2L-$Lipschitz Hessian.

Applying (1) and 14 to $f_i - f$ we get

$$\|\mathcal{G}(f_i, \boldsymbol{x}, \tilde{\boldsymbol{x}}) - \nabla f(\boldsymbol{x})\| \le L\|\boldsymbol{x} - \tilde{\boldsymbol{x}}\|^2$$

and

$$\|\mathcal{H}(f_i, \boldsymbol{x}, \tilde{\boldsymbol{x}}) - \nabla^2 f(\boldsymbol{x})\| \le 2L\|\boldsymbol{x} - \tilde{\boldsymbol{x}}\|$$

We note also the if $i$ is chosen at random then $\mathbb{E}_i\mathcal{G}(f_i, \boldsymbol{x}, \tilde{\boldsymbol{x}}) = \nabla f(\boldsymbol{x})$ and $\mathbb{E}_i\mathcal{H}(f_i, \boldsymbol{x}, \tilde{\boldsymbol{x}}) = \nabla^2 f(\boldsymbol{x})$.

By using the properties of variance we have

$$\mathbb{E}_{\mathcal{B}}\|\mathcal{G}(f_{\mathcal{B}}, \boldsymbol{x}, \tilde{\boldsymbol{x}}) - \nabla f(\boldsymbol{x})\|^2 \le \frac{\mathbb{E}_i\|\mathcal{G}(f_i, \boldsymbol{x}, \tilde{\boldsymbol{x}}) - \nabla f(\boldsymbol{x})\|^2}{b} \le \frac{L^2}{b}\|\boldsymbol{x} - \tilde{\boldsymbol{x}}\|^4$$

Now all that is left is to apply Lemmas 6 and 7 we get

$$\mathbb{E}_{\mathcal{B}}\|\mathcal{G}(f_{\mathcal{B}}, \boldsymbol{x}, \tilde{\boldsymbol{x}}) - \nabla f(\boldsymbol{x})\|^3 \le \frac{L^3}{b^{3/2}}\|\boldsymbol{x} - \tilde{\boldsymbol{x}}\|^3$$

And

$$\mathbb{E}_{\mathcal{B}}\|\mathcal{H}(f_{\mathcal{B}}, \boldsymbol{x}, \tilde{\boldsymbol{x}}) - \nabla^2 f(\boldsymbol{x})\|^3 \le \left(2\sqrt{\frac{2e\log(d)}{b}} + \frac{8e\log(d)}{b}\right)^3 \mathbb{E}_i\|\mathcal{H}(f_i, \boldsymbol{x}, \tilde{\boldsymbol{x}}) - \nabla^2 f(\boldsymbol{x})\|^3$$

$$\le \left(2\sqrt{\frac{2e\log(d)}{b}} + \frac{8e\log(d)}{b}\right)^3 L^3\|\boldsymbol{x} - \tilde{\boldsymbol{x}}\|^3$$

## C.2   Proof of Theorem 2

We use Theorem 5 and denote denoting $r_{i+1} = \|\boldsymbol{x}_{i+1} - \boldsymbol{x}_i\|$ then by the definition of the similarity of $h_1, h_2$ to $f$ we have:

$$\mathbb{E}f(\boldsymbol{x}_{sm+i}) - \mathbb{E}f(\boldsymbol{x}_{sm+i+1}) \ge \frac{1}{216\sqrt{M}}\mathbb{E}\mu_M(\boldsymbol{x}_{sm+i+1}) + \mathbb{E}\left[\frac{M}{72}r_{sm+i+1}^3 - \left(\frac{4\delta_1^{3/2}}{\sqrt{M}} + \frac{73\delta_2^3}{M^2}\right)\|\boldsymbol{x}_{sm+i} - \boldsymbol{x}_{sm}\|^3\right]$$

We sum from $i = 0$ to $i = m - 1$

$$\mathbb{E}f(x_{sm}) - \mathbb{E}f(x_{(s+1)m}) \ge \sum_{i=0}^{m-1}\frac{1}{216\sqrt{M}}\mathbb{E}\mu_M(\boldsymbol{x}_{sm+i+1}) + \mathbb{E}\left[\frac{M}{72}r_{sm+i+1}^3 - \left(\frac{4\delta_1^{3/2}}{\sqrt{M}} + \frac{73\delta_2^3}{M^2}\right)\|\boldsymbol{x}_{sm+i} - \boldsymbol{x}_{sm}\|^3\right]$$

We note that $\|\boldsymbol{x}_{sm+i} - \boldsymbol{x}_{sm}\| \le \sum_{j=1}^{i-1}r_{sm+j}$, this means

$$\mathbb{E}f(x_{sm}) - \mathbb{E}f(x_{(s+1)m}) \ge \sum_{i=0}^{m-1}\frac{1}{216\sqrt{M}}\mathbb{E}\mu_M(\boldsymbol{x}_{sm+i+1}) + \mathbb{E}\left[\frac{M}{72}r_{sm+i+1}^3 - \left(\frac{4\delta_1^{3/2}}{\sqrt{M}} + \frac{73\delta_2^3}{M^2}\right)\left(\sum_{j=1}^{i-1}r_{sm+j}\right)^3\right]$$

We apply now the following inequality (from [10]) $\sum_{k=1}^{m-1}\left(\sum_{i=1}^{k}r_i\right)^3 \le \frac{m^3}{3}\sum_{k=1}^{m-1}r_k^3$ true for positive

numbers $\{r_k\}_{k\ge 1}$ and any $m \ge 1$. This inequality means :

$$\sum_{i=0}^{m-1}\left[\frac{M}{72}r_{sm+i+1}^3 - \left(\frac{4\delta_1^{3/2}}{\sqrt{M}} + \frac{73\delta_2^3}{M^2}\right)\left(\sum_{j=1}^{i-1}r_{sm+j}\right)^3\right] \ge \left(\frac{M}{72} - \frac{m^3}{3}\left(\frac{4\delta_1^{3/2}}{\sqrt{M}} + \frac{73\delta_2^3}{M^2}\right)\right)\sum_{i=0}^{m-1}r_{sm+i+1}^3$$

478    The above quantity is thus positive if $\frac{M}{72} - \frac{m^3}{3}\left(\frac{4\delta_1^{3/2}}{\sqrt{M}} + \frac{73\delta_2^3}{M^2}\right) \geq 0$.

479    Equivalently, for $M$ satisfying

$$4(\tfrac{\delta_1}{M})^{3/2} + 73(\tfrac{\delta_2}{M})^3 \leq \tfrac{1}{24m^3} \tag{24}$$

We have:

$$\mathbb{E}f(x_{sm}) - \mathbb{E}f(x_{(s+1)m}) \geq \tfrac{m}{216\sqrt{M}} \tfrac{1}{m} \sum_{i=0}^{m-1} \mathbb{E}\mu(x_{sm+i+1}).$$

We sum from $s = 0$ to $s = S - 1$ which gives :

$$\frac{1}{216Sm} \sum_{s=0}^{S-1} \sum_{i=0}^{m-1} \mathbb{E}\mu(x_{sm+i+1}) \leq \frac{\sqrt{M}(f(x_0) - f^\star)}{Sm}$$

And thus by definition of $\boldsymbol{x}_{out}$ we have:

$$\mathbb{E}\mu(x_{out}) \leq 216 \frac{\sqrt{M}(f(x_0) - f^\star)}{Sm}$$

480   # D   Gradient dominated functions

481   ## D.1   Examples of gradient-dominated functions

482    Let us provide several main examples of functions satisfying (11):

> **Example 1** *Let $f$ be convex on a bounded convex set $Q$ of diameter $D$, and let solution $\boldsymbol{x}^\star$ to*
> (1) *belong to $Q$. Then, we have:*
>
> $$f(\boldsymbol{x}) - f^\star \quad \leq \quad \langle \nabla f(\boldsymbol{x}), \boldsymbol{x} - \boldsymbol{x}^\star \rangle \quad \leq \quad D\|\nabla f(\boldsymbol{x})\|, \qquad \forall \boldsymbol{x} \in Q.$$
>
> *Therefore, $f$ is $(D, 1)$-gradient dominated.*

> **Example 2** *Let $f$ be uniformly convex of degree $s \geq 2$ with some constant $\sigma > 0$:*
>
> $$f(\boldsymbol{y}) \quad \geq \quad f(\boldsymbol{x}) + \langle \nabla f(\boldsymbol{x}), \boldsymbol{y} - \boldsymbol{x} \rangle + \tfrac{\sigma}{s}\|\boldsymbol{y} - \boldsymbol{x}\|^s, \qquad \forall \boldsymbol{x}, \boldsymbol{y} \in \mathbb{R}^d.$$
>
> *Then, $f$ is $\left(\frac{s-1}{s}(\frac{1}{\sigma})^{\frac{1}{s-1}}, \frac{s}{s-1}\right)$-gradient dominated (see, e.g. [11]).*

483    In particular, uniformly convex functions of degree $s = 2$ are known as *strongly convex*, and we see
484    that they satisfy condition (11) with $\tau = \frac{1}{2\sigma}$ and $\alpha = 2$. However, the function class (11) is much
485    wider and it includes also some problems with *non-convex objectives* ([21]).

486   ## D.2   Special cases of Theorem 3

487    For **Convex functions.** Theorem 3 implies that for $M = \max\left(L, \frac{\sigma_h T}{2D}\right)$ we have the rate

$$\mathbb{E}[f(\boldsymbol{x}_{out})] - f^\star = \mathcal{O}\left(\frac{LD^3}{T^2} + \frac{\sigma_h D^2}{T} + \sigma_g D\right). \tag{25}$$

488    Equation (25) has been obtained by [1] but under the much stronger assumption of almost surely
489    bounded noise. Using the gradient and Hessian estimates in (23), for $\varepsilon > 0$ and $M = L$, to reach an
490    $\varepsilon$-global minimum, we need at most $T = \mathcal{O}(\sqrt{\frac{LD^3}{\varepsilon}})$, $b_h = \mathcal{O}(\frac{\sigma_h^2 D}{L\varepsilon})$ and $b_g = \mathcal{O}(\frac{\sigma_g^2 D^2}{\varepsilon^2})$. In other
491    words, we need at most $\mathcal{O}(\frac{\sigma_g^2 L^{1/2} D^{3/2}}{\varepsilon^{5/2}} + d\frac{\sigma_h^2 D^{5/2}}{L^{1/2}\varepsilon^{3/2}})$   *GradCost*.
492    **s-uniformly convex functions.** For this class of functions, using the estimates in (23) and for $\varepsilon > 0$
493    and $M = L$, we reach an $\varepsilon$-global minimum in at most $T = \mathcal{O}(\frac{\sqrt{L}}{s^2}\log(\frac{F_0}{\varepsilon}))$, $b_h = \mathcal{O}(\frac{\sigma_h^2}{s^{4/3}L\varepsilon^{2/3}})$
494    and $b_g = \mathcal{O}(\frac{\sigma_g^2}{s^{8/3}\varepsilon^{4/3}})$ or equivalently $\tilde{\mathcal{O}}(\frac{\sigma_g^2\sqrt{L}}{s^{14/3}\varepsilon^{4/3}} + d\frac{\sigma_h^2}{s^{10/3}\sqrt{L}\varepsilon^{2/3}})$ *GradCost*.
495    $\mu$-**strongly convex functions.** For this class of functions, for $M = L$, for any $\varepsilon > 0$ to get
496    $\mathbb{E}[f(\boldsymbol{x}_{out})] - f^\star < \varepsilon$ we need at most $T = \mathcal{O}(t_0 + \log\log(\frac{\mu^3}{L^2\varepsilon}))$, $b_h = \mathcal{O}(\frac{\sigma_h^2}{L\sqrt{\mu\varepsilon}})$ and $b_g = \mathcal{O}(\frac{\sigma_g^2}{\mu\varepsilon})$
497    or $\tilde{\mathcal{O}}(\frac{t_0\sigma_g^2}{\mu\varepsilon} + d\frac{t_0\sigma_h^2}{L\sqrt{\mu\varepsilon}})$ *GradCost*.

## D.3 A special case of Theorem 4

Since we have here many special cases depending on the value of $\alpha$ and the choices of the helpers, we will only consider the case of **convex functions** i.e. $\alpha = 1, \tau = D$, but it is easy to apply Theorem 4 to other cases (like uniformly convex and strongly convex functions).

Theorem 4 implies for convex functions the following:

$$\mathbb{E}[f(\boldsymbol{x}_{out})] - f^\star = \mathcal{O}\left(\frac{\delta_1 D^3}{S^2} + \frac{\delta_2 D^3}{S^2 m} + \frac{LD^3}{S^2 m^2}\right)$$

We have the following special cases based on the choice of the helper functions:

- Cubic Newton [21] corresponds to $\delta_1 = \delta_2 = 0$, we get indeed its known rate in the convex case. Under the **SOGEO** oracle, we reach an $\varepsilon$-global minimum in at most $\mathcal{O}(\frac{nd}{\sqrt{\varepsilon}})$ *GradCost*.
- Convex Lazy Cubic Newton (which was not considered in [10]) corresponds to $\delta_1 = 0, \delta_2 = L$, which gives $\mathbb{E}[f(\boldsymbol{x}_{out})] - f^\star = \mathcal{O}\left(\frac{LD^3}{S^2 m}\right)$. Under the **SOGEO** oracle, by choosing $m = d$ we reach an $\varepsilon$-global minimum in at most $\mathcal{O}(\frac{n\sqrt{d}}{\sqrt{\varepsilon}})$ *GradCost*.
- Convex variance reduced cubic Newton (also not considered by [36, 30]), corresponds to each time sampling $\mathcal{B}_g, \mathcal{B}_h$ of sizes $b_g, b_h$ consecutively at random and setting $h_1 = \frac{1}{b_g} \sum_{i \in \mathcal{B}_g} f_i, h_2 = \frac{1}{b_h} \sum_{i \in \mathcal{B}_g} f_i$. According to Lemma 1 we have $\delta_1 = \frac{L}{\sqrt{b_g}}$ and $\delta_2 = \tilde{\mathcal{O}}(\frac{L}{\sqrt{b_h}})$ so for $b_g \sim m^4, b_h \sim m^2$ and $M = L$ we get $\mathbb{E}[f(\boldsymbol{x}_{out})] - f^\star = \mathcal{O}\left(\frac{LD^3}{S^2 m^2}\right)$. Again, Under the **SOGEO** oracle, by choosing $m = (nd)^{1/5} 1_{d \leq n^{2/3}} + n^{1/3} 1_{d \geq n^{2/3}}$, we reach an $\varepsilon$-global-minimum in at most $\mathcal{O}\left(\frac{\min\left((nd)^{4/5}, n^{2/3} d + n\right)}{\sqrt{\varepsilon}}\right)$ *GradCost*.
- Variance reduced cubic newton with Lazy Hessians, in this case, by using sampling, we have $\delta_1 = \frac{L}{\sqrt{b_g}}$ and $\delta_2 = L$. If we take $m = (nd)^{1/3} 1_{d \leq \sqrt{n}} + d 1_{d \geq \sqrt{n}}$ then we reach an $\varepsilon$-global-minimum in at most $\mathcal{O}\left(\frac{\min\left((nd)^{5/6}, n\sqrt{d}\right)}{\sqrt{\varepsilon}}\right)$ *GradCost*. Again this improves the complexity of Lazy Cubic Newton.

## D.4 Proof of Theorem 3

From the previous proof we have

$$\mathbb{E}f(\boldsymbol{x}_t) - \mathbb{E}f(\boldsymbol{x}_{t+1}) \geq \frac{1}{1008\sqrt{M}}\mathbb{E}\|\nabla f(\boldsymbol{x}_{t+1})\|^{3/2} - \frac{4}{\sqrt{M}}\sigma_g^{3/2} - \frac{73}{M^2}\sigma_h^3$$

By the definition of $(\tau, \alpha)$-gradient dominated functions we have

$$f(\boldsymbol{x}_t) - f^\star \leq \tau\|\nabla f(\boldsymbol{x}_t)\|^\alpha$$

So

$$\mathbb{E}\|\nabla f(\boldsymbol{x}_{t+1})\|^{3/2} \geq \mathbb{E}\big(\frac{f(\boldsymbol{x}_{t+1}) - f^\star}{\tau}\big)^{\frac{3}{2\alpha}}.$$

If $\alpha \leq 3/2$, then by Jensen's inequality we have

$$\mathbb{E}\|\nabla f(\boldsymbol{x}_{t+1})\|^{3/2} \geq \big(\frac{\mathbb{E}f(\boldsymbol{x}_{t+1}) - f^\star}{\tau}\big)^{\frac{3}{2\alpha}}.$$

For $\alpha > 3/2$ we need to assume that $\mathbb{E}f(\boldsymbol{x}_t) - f^\star \leq \tau\mathbb{E}[\|\nabla f(\boldsymbol{x}_t)\|]^\alpha$ which will give us also $\mathbb{E}\|\nabla f(\boldsymbol{x}_{t+1})\|^{3/2} \geq \big(\frac{\mathbb{E}f(\boldsymbol{x}_{t+1}) - f^\star}{\tau}\big)^{\frac{3}{2\alpha}}$.

We will consider the sequence $F_t = \mathbb{E}f(\boldsymbol{x}_t) - f^\star$ and denote $\gamma = \frac{3}{2\alpha}$, $C = \frac{1}{1008\sqrt{M}\tau^\gamma}$ and $a = \frac{4}{\sqrt{M}}\sigma_g^{3/2} + \frac{25}{M^2}\sigma_h^3$. Then the sequence $(F_t)$ satisfies :

$$F_t - F_{t+1} \geq CF_{t+1}^{\gamma} - a \tag{26}$$

- Case $\gamma = 1$ then

$$F_{t+1} \geq \frac{F_t + a}{C + 1}$$

So by recurrence, we have:

$$F_t \leq (1 + C)^{-t} F_0 + \sum_{i=0}^{t-1} (1 + C)^{-i} \frac{a}{1 + C} \leq (1 + C)^{-t} F_0 + \frac{a}{C}$$

We note that $(1 + C)^{-t} \leq \exp(\frac{-Ct}{1+C})$ So

$$F_t \leq \exp(\frac{-Ct}{1 + C}) F_0 + \frac{a}{C}$$

- Case $\gamma \in (1, 2]$ then let $\tilde{F}_t = \frac{F_t}{C^{1/(1-\gamma)}}$ and $\tilde{a} = \frac{a}{C^{1/(1-\gamma)}}$ for which we have

$$\tilde{F}_t - \tilde{F}_{t+1} \geq \tilde{F}_{t+1}^{\gamma} - \tilde{a}$$

Now let $x = \tilde{a}^{1/\gamma}$, and $\delta_t = \tilde{F}_t - x$ so :

$$\delta_t - \delta_{t+1} \geq (\delta_{t+1} + x)^{\gamma} - x^{\gamma} \geq \delta_{t+1}^{\gamma}$$

Where we used in the last inequality the fact that $(x + y)^{\gamma} \geq x^{\gamma} + y^{\gamma}$ for $\gamma \geq 1$ and $x, y \geq 0$.

All in all $\delta_t = \frac{F_t - (\frac{a}{C})^{1/\gamma}}{C^{1/(1-\gamma)}}$ and

$$\delta_t - \delta_{t+1} \geq \delta_{t+1}^{\gamma}$$

If $\delta_{t+1} \geq \delta_t/2$ then

$$\frac{1}{(\gamma - 1)\delta_{t+1}^{\gamma-1}} - \frac{1}{(\gamma - 1)\delta_t^{\gamma-1}} \geq \frac{\delta_t^{\gamma-1} - \delta_{t+1}^{\gamma-1}}{(\gamma - 1)\delta_t^{\gamma-1}\delta_{t+1}^{\gamma-1}}$$

By concavity of $x \mapsto x^{\gamma-1}$ (since $\gamma \leq 2$) we get :

$$\frac{1}{(\gamma - 1)\delta_{t+1}^{\gamma-1}} - \frac{1}{(\gamma - 1)\delta_t^{\gamma-1}} \geq \frac{\delta_t - \delta_{t+1}}{(\gamma - 1)\delta_t\delta_{t+1}^{\gamma-1}} \geq \frac{\delta_{t+1}}{\delta_t} \geq 1/2$$

If $\delta_{t+1} \leq \delta_t/2$ then

$$\frac{1}{(\gamma - 1)\delta_{t+1}^{\gamma-1}} - \frac{1}{(\gamma - 1)\delta_t^{\gamma-1}} \geq \frac{1}{(\gamma - 1)\delta_t^{\gamma-1}}(2^{\gamma-1} - 1) \geq \frac{2^{\gamma-1} - 1}{(\gamma - 1)\delta_0^{\gamma-1}}$$

Using the fact that $(\delta_t)$ is decreasing.

In all cases we have:

$$\frac{1}{(\gamma - 1)\delta_{t+1}^{\gamma-1}} - \frac{1}{(\gamma - 1)\delta_t^{\gamma-1}} \geq max(1/2, \frac{2^{\gamma-1} - 1}{(\gamma - 1)\delta_0^{\gamma-1}}) := D$$

By summing from $t = 0$ to $t = T - 1$ we get :

$$\frac{1}{(\gamma - 1)\delta_T^{\gamma-1}} \geq DT$$

In other words

$$\delta_T \leq (\frac{1}{(\gamma - 1)DT})^{\frac{1}{\gamma-1}}$$

- Case $\gamma < 1$: then we have

$$F_{t+1} \le \big(\frac{F_t - F_{t+1} + a}{C}\big)^{1/\gamma}$$

By convexity of $x \mapsto x^{1/\gamma}$ we get

$$F_{t+1} \le 2^{1/\gamma - 1}\big(\frac{F_t - F_{t+1}}{C}\big)^{1/\gamma} + 2^{1/\gamma - 1}\big(\frac{a}{C}\big)^{1/\gamma}$$

Let $\delta_t = \frac{F_t - 2^{1/\gamma - 1}(\frac{a}{C})^{1/\gamma}}{2^{1/\gamma}C^{1/(1-\gamma)}}$ then we have

$$\delta_{t+1} \le (\delta_t - \delta_{t+1})^{1/\gamma}$$

The sequence $(\delta_t)$ is decreasing thus $\delta_{t+1} \le \delta_t^{1/\gamma}$ which guarantees a superlinear rate the moment $\delta_t < 1$.

In fact, we can show that at the beginning $(\delta_t)$ will decrease at least at a linear rate, and thus it will be at some point $< 1$.

We have $\frac{\delta_t}{\delta_{t+1}} \ge 1 + \frac{\delta_t - \delta_{t+1}}{\delta_{t+1}} \ge 1 + \frac{1}{\delta_{t+1}^{1-\gamma}} \ge 1 + \frac{1}{\delta_0^{1-\gamma}}$

Which means $\delta_{t+1} \le (1 + \frac{1}{\delta_0^{1-\gamma}})^{-1}\delta_t = (1 - \frac{1}{1+\delta_0^{1-\gamma}})\delta_t \le \exp(-\frac{1}{1+\delta_0^{1-\gamma}})\delta_t$.

To have $\delta_t \le 1/2$ we need $t \ge t_0 = (1 + \delta_0^{1-\gamma})\log(2\delta_0)$ so that for $t \ge t_0$ we enjoy a superlinear rate and we have $\delta_t \le \big(\frac{1}{2}\big)^{(\frac{1}{\gamma})^{t-t_0}}$

This finishes the proof.

## D.5 Proof of Theorem 4

In Theorem 4 we made the choice of updating the snapshot in the following way $\tilde{\boldsymbol{x}}_{s+1} = \boldsymbol{x}_{\arg\min_{i\in\{0,\cdots,m-1\}} f(\boldsymbol{x}_{sm+i})}$ which means that $f(\tilde{\boldsymbol{x}}_{s+1}) \le f(\boldsymbol{x}_{sm+i})$ for all $i \in \{0,\cdots,m-1\}$.

For $s \in \{0,\cdots,S-1\}$ and $i \in \{0,\cdots,m-1\}$ We have the following inequality

$$\mathbb{E}f(\boldsymbol{x}_{sm+i}) - \mathbb{E}f(\boldsymbol{x}_{sm+i+1}) \ge \frac{1}{216\sqrt{M}}\mathbb{E}\|\nabla f(\boldsymbol{x}_{sm+i+1})\|^{3/2} + \mathbb{E}\Big[\frac{M}{72}r_{sm+i+1}^3 - \big(\frac{4\delta_1^{3/2}}{\sqrt{M}} + \frac{73\delta_2^3}{M^2}\big)\|\boldsymbol{x}_{sm+i} - \boldsymbol{x}_{sm}\|^3\Big]$$

By definition of gradient-dominated functions we have $\mathbb{E}\|\nabla f(\boldsymbol{x}_{t+1})\|^{3/2} \ge \big(\frac{\mathbb{E}f(\boldsymbol{x}_{t+1}) - f^\star}{\tau}\big)^{\frac{3}{2\alpha}}$.

So

$$\mathbb{E}f(\boldsymbol{x}_{sm+i}) - \mathbb{E}f(\boldsymbol{x}_{sm+i+1}) \ge \frac{1}{216\sqrt{M}}\big(\frac{\mathbb{E}f(\boldsymbol{x}_{sm+i+1}) - f^\star}{\tau}\big)^{\frac{3}{2\alpha}}$$
$$+ \mathbb{E}\Big[\frac{M}{72}r_{sm+i+1}^3 - \big(\frac{4\delta_1^{3/2}}{\sqrt{M}} + \frac{73\delta_2^3}{M^2}\big)\|\boldsymbol{x}_{sm+i} - \boldsymbol{x}_{sm}\|^3\Big]$$
$$\ge \frac{1}{216\sqrt{M}}\big(\frac{\mathbb{E}f(\tilde{\boldsymbol{x}}_{s+1}) - f^\star}{\tau}\big)^{\frac{3}{2\alpha}} + \mathbb{E}\Big[\frac{M}{72}r_{sm+i+1}^3 - \big(\frac{4\delta_1^{3/2}}{\sqrt{M}} + \frac{73\delta_2^3}{M^2}\big)\|\boldsymbol{x}_{sm+i} - \boldsymbol{x}_{sm}\|^3\Big]$$

Summing the above inequality from $i = 0$ to $i = m - 1$ and remarking that $\tilde{\boldsymbol{x}}_s = \boldsymbol{x}_{sm}$ we get

$$\mathbb{E}f(\tilde{\boldsymbol{x}}_s) - \mathbb{E}f(\boldsymbol{x}_{(s+1)m}) \ge \frac{m}{216\sqrt{M}}\big(\frac{\mathbb{E}f(\tilde{\boldsymbol{x}}_{s+1}) - f^\star}{\tau}\big)^{\frac{3}{2\alpha}} + \mathbb{E}\Big[\sum_{i=0}^{m-1}\frac{M}{72}r_{sm+i+1}^3 - \big(\frac{4\delta_1^{3/2}}{\sqrt{M}} + \frac{73\delta_2^3}{M^2}\big)\|\boldsymbol{x}_{sm+i} - \boldsymbol{x}_{sm}\|^3\Big]$$

By definition of $\tilde{\boldsymbol{x}}_{s+1}$ we have $f(\tilde{\boldsymbol{x}}_{s+1}) \le f(\tilde{\boldsymbol{x}}_{sm+i})$ for all $i \in \{0,\ldots,m-1\}$ which leads to

$$\mathbb{E}f(\tilde{\boldsymbol{x}}_s) - \mathbb{E}f(\tilde{\boldsymbol{x}}_{s+1}) \ge \frac{m}{216\sqrt{M}}\big(\frac{\mathbb{E}f(\tilde{\boldsymbol{x}}_{s+1}) - f^\star}{\tau}\big)^{\frac{3}{2\alpha}} + \mathbb{E}\Big[\sum_{i=0}^{m-1}\frac{M}{72}r_{sm+i+1}^3 - \big(\frac{4\delta_1^{3/2}}{\sqrt{M}} + \frac{73\delta_2^3}{M^2}\big)\|\boldsymbol{x}_{sm+i} - \boldsymbol{x}_{sm}\|^3\Big]$$

For $M$ satisfying (24) we have $\sum_{i=0}^{m-1} \frac{M}{72} r_{sm+i+1}^3 - \left( \frac{4\delta_1^{3/2}}{\sqrt{M}} + \frac{73\delta_2^3}{M^2} \right) \|\boldsymbol{x}_{sm+i} - \boldsymbol{x}_{sm}\|^3 \geq 0$ thus we have

$$\mathbb{E}f(\tilde{\boldsymbol{x}}_s) - \mathbb{E}f(\tilde{\boldsymbol{x}}_{s+1}) \geq \frac{m}{216\sqrt{M}} \left( \frac{\mathbb{E}f(\tilde{\boldsymbol{x}}_{s+1}) - f^\star}{\tau} \right)^{\frac{3}{2\alpha}}.$$

Let's define $F_s = \mathbb{E}f(\tilde{\boldsymbol{x}}_s) - f^\star$ then

$$F_s - F_{s+1} \geq C F_{s+1}^\gamma$$

which is a special case of the sequence 26 with $a = 0$, thus we can apply our findings from before and replace $a$ by 0.