# OpenReview forum: "Unified Convergence Theory of Stochastic and Variance-Reduced Cubic Newton Methods"
_NeurIPS.cc/2023/Conference — Submitted to NeurIPS 2023_

### Official Review · Reviewer_fBtY · 2023-06-22

**Soundness:** 3 good
**Presentation:** 4 excellent
**Contribution:** 3 good
**Rating:** 8
**Confidence:** 3

**Summary:**

The authors propose a new way of solving minimization problems that are not necessarily convex.
The main element of their approach is that they use some helper functions which give auxiliary information about the optimization problem at hand.
That enables them to give improved optimization algorithms.

**Strengths:**

Originality:
The main approach is original.
That is, combining stochastic and variance-induced second-order algorithms is a novel direction.

Quality:
Writing and results are of good quality; see questions.

Clarity:
Clear writing; see questions.

Significance:
Work is significant, as optimization is a central task in computer science.

**Weaknesses:**

Not significant weaknesses found;
see questions.

**Questions:**

Line 28:
Why is a saddle point not desirable?

Line 30:
Please explain what you mean by ill-conditioned.

Line 58:
What is the PL condition?

Line 60:
Can you please elaborate on the discussion regarding previous work?

Line 70:
Unclear.

Line 83:
Bullet is unclear.
Why is it important in the parameter is smaller?

Line 86:
Bullet:
Why are stochastic Cubic Newton algorithms with variance reduction for gradient-dominated case important?
Because of their applicability?

Line 98:
Can you please discuss a bit Assumption 1?
It is not trivial, right?

Line 105:
Can you please elaborate on the accuracy measure $\mu_c$?

Line 120:
Can you please cite some relevant work here?

Line 157:
Can you please elaborate on Assumption 2?

Line 271:
I do not understand the discussion before Equation (12).

**Limitations:**

Yes.

---

> ### Author Rebuttal · Authors · 2023-08-09
>
> Thank you very much for the invested time and expertise, and for the positive feedback as well. We hope that all raised issues are properly addressed in our rebuttal.
>
>
> - **Line 28:** A saddle point would not be desirable if our goal is to find a minimum, finding a saddle point means that we missed a better local minimum. To be fair, not all saddle points are worse than all local minimums, but to each saddle point, there is a better local minimum, given that the objective function is lower bounded. There is also a belief in the ML community that saddle points encountered in practical ML settings are bad, for example in an [article](https://arxiv.org/abs/1406.2572) by Yoshua Bengio et al there is the following conjecture:
> >[...]a deeper and more profound difficulty originates from the proliferation of saddle points, not local minima, especially in high dimensional problems of practical interest. Such saddle points are surrounded by high error plateaus that can dramatically slow down learning, and give the illusory impression of the existence of a local minimum.
>
> - **Line 30:** A problem is ill-conditioned when a small relative error in its inputs (parameters) can cause a large relative error in its outputs. In optimization theory, this is usually measured by the condition number. The main conclusion is that the more ill-conditioned our problem is the more steps we need, and thus using a faster method might be desirable.
>
> - **Line 58:** The $\mu$-PL condition is $ f(x) - f(x^\star) \leq \frac{1}{2\mu}\|\nabla f(x)\|^2$. It is a special case of gradient-dominated functions that is implied by strong convexity and was shown to hold for some special cases of neural networks, see Lines 268-269.
>
> - **Line 60:** we discuss relevant works in lines 44-59. Basically, we use the Cubic Newton method as the base algorithm, introduced by Nesterov and Polyak [20], a recent Lazy Cubic Newton method [9] (Line 50), stochastic and subsampled Cubic Newton methods (Line 44) which use batches to approximate both gradients and Hessians, and the variance-reduced stochastic Cubic Newton methods (Line 47). There are also several works that study the convergence of such algorithms for gradient-dominated functions (Line 55). In our paper, we unify the analysis of all of these algorithms. As a direct consequence of our theory, we obtain improved complexity guarantees for variance-reduced algorrithms and propose a new method (VRCN-Lazy) with improved global complexities (please, see also the table in attached PDF).
>
> - **Line 83:** Why replacing the smoothness constant by the similarity is a good thing? The complexity of Cubic Newton is proportional to $\sqrt{L}$ so if the similarity $\delta$ is small and we replace $L$ with it then we get a smaller overall complexity.
>
> - **Line 86:** Gradient-dominated functions are important because although they might be non-convex, we prove that they converge to a global minimum, usually much faster, such functions include convex functions, functions satisfying PL condition, and some neural networks.
>
> - **Line 98:** Assumption 1 roughly means that the Hessians at two close points should be close, how close they are is bounded by how close the points are. This is the equivalent of the smoothness assumption for first-order methods.
>
> - **Line 105:** The accuracy measure $\mu_c$ is small if both the gradient norm is small and the smallest Eigenvalue of the Hessian is nearly positive, in other words when it is small this means we are close to a local minimum (not a saddle point).
>
> - **Line 157:** Assumptions 2 and 3 assume that the gradient $\mathcal{G}$ and the Hessian $\mathcal{H}$ are close to those of the main function.
>
> - **Line 120:** Relevant work. there is the MM algorithm by Lange [MM](https://www.stat.berkeley.edu/~aldous/Colloq/lange-talk.pdf), MM in the context of minimization stands for majorize-minimize, basically to majorize a function and then minimize the majorization instead of the function. This is usually stressed for the EM algorithm, but it applies more generally. Usually, this principle is hidden but most optimization algorithms implicitly use it.

---

### Official Review · Reviewer_bq1t · 2023-07-06

**Soundness:** 3 good
**Presentation:** 3 good
**Contribution:** 2 fair
**Rating:** 5
**Confidence:** 4

**Summary:**

In this paper, the authors introduce a meta-algorithm for second-order methods which they refer to as the ``helper framework''. Their helper framework unifies stochastic and variance-reduced versions of Newton's method, and the framework could also be viewed as auxiliary tasks when applied to first-order methods. The paper uses the helper framework to establish various convergence rates for objectives with Lipschitz Hessian and helper functions with two different similarity conditions (bounded similarity and Lipschitz similarity). Their algorithm (Algorithm 1) applies this framework to analyze cubic Newton. Different settings of Algorithm 1 (e.g. using different helper functions) replicate stochastic Newton, variance-reduced Newton, stochastic and variance-reduced Newton, lazy Hessian Newton, etc. They apply the helper framework to gradient-dominated objectives. They derive convergence rates for stochastic cubic Newton algorithms with variance reduction in the gradient-dominated case, which they claim to be the first to establish.

**Strengths:**

1. Introduces a new framework to unify analysis of different variants of Newton's method.
2. Instead of assuming convexity or strong-convexity, extends framework to gradient-dominated functions in general.
3. Through this framework, the authors claim to achieve improved complexity guarantees.
4. Well-written paper with clear notation and good recap of related work.

**Weaknesses:**

1. Unclear why we need a new framework to analyze the different variants of Newton's method.
2. Unclear if the meta-algorithm presented is implementable.
3. The improved complexity guarantees claimed in the paper are unclear.
4. The additional variables in the helper framework may make it harder to understand basic second-order methods.

The reason why I only rated a 2 for Contribution of this paper are mainly due to the four points above. I may have misread or overlooked some details. Please see the questions below that address the perceived weaknesses in more detail.

**Questions:**

Questions:
1. Is it useful to have a framework that unifies the different variants of Newton's method? What additional insights are revealed by using the helper framework?
2. Could you expand on the statement ``we recuperate the known rates of convergence'' on line 213? For readers unfamiliar with the convergence rates of various second-order methods, it could be useful to have a table that shows (method, previous known rate, rate from helper framework).
3. Is their main algorithm (Algorithm 1) implementable/ practical? For example, in one case (line 208), the input parameter $m$ in Algorithm 1 is value that minimizes the number of gradient and Hessian evaluations to find an $\epsilon$ stationary point. This seems to be a difficult optimization problem itself. Are there more practical ways of setting $m$? Another example is in Corollary 1 where $M$ is chosen to equal $L$.
4. The framework introduces some variables such as $\delta_1$ and $\delta_2$ in Assumptions 2 and 3. This is expected when a more general framework is introduced and we have additional concepts such as similarity. On the other hand, there is no closed-form for the values of these new variables in general and they also appear hard to estimate. Could the authors provide more insight/ theory/ experimental results for the new variables introduced so that the new complexity guarantees are more understandable?

Minor questions:
1. The parameter $M$ is used in equation 2 but not defined until later at line 123?
2. What is $F_0$ in Theorem 1?
3. Line 169. Could you be more specific about which inequality is used so that Assumption 2 is satisfied?
4. Line 172. Is the result of Corollary 1 a complexity guarantee for a second-order method? If so, why is the comparison against first-order SGD?
5. Is there any difference between the definitions of $h_1$ and $h_2$ on line 215 and Equation 6?
6. There seems to be experimental results for the discussion of ``To be lazy or not to be?'' in the appendix. Why not add a link to this?
7. Line 244. The similarity measures in Assumption 2 and 3 only take in $\delta_1$ and $\delta_2$. What is the second variable in $(\delta_1,1)$ and $(\delta_2,2)$?


**Limitations:**

Yes

---

> ### Author Rebuttal · Authors · 2023-08-09
>
> Thank you very much for the invested time and expertise, and for the positive feedback as well. We hope that all raised issues are properly addressed in our rebuttal.
>
> 1. **Helper framework.** Using a unified framework is always beneficial because it helps us see different approaches as connected. In our context, this involves more than just consolidating existing variants of the Newton Method with Cubic Regularization. Our theory encompasses Lazy methods as well, which require fewer Hessian evaluations. Moreover, our approach offers a broader perspective that extends to Auxiliary learning and semi-supervised learning.
>
> 2. **Convergence Rates.** Please, find attached a PDF file with a new table with global complexities, which we also include into the main part of our paper. In this table, we compare our results with stochastic and variance-reduced first-order and second-order methods.
>
>
> 3. **Implementation.**  Our Algorithm 1 is efficiently implementable in practice. We apologize for a limited set of concrete examples due to the lack of space. Our general frameworks cover existing stochastic second-order methods and the methods with variance reduction that are known in the previous literature. Moreover, based on our framework, we propose a new stochastic second-order method (VRCN-Lazy) with improved global complexities (please, see also the table in attached PDF), that requires less computational efforts for solving the problem with a given accuracy.
>
> The choice of our key parameter $m$ (the frequency of snapshot updates) is actually possible in practice. In the paper, we provide appropriate choices for $m$ in several relevant special cases. In general, it only depends on the number of samples $n$ and the dimension $d$, and it needs to optimize the total arithmetic complexity of the computations, given the known cost of one stochastic gradient / Hessian computation. For example, in the important particular case $n = 1$ (minimizing just one non-convex function), the optimal choice is $m = d$ (update the Hessian once per $d$ steps of the method, where $d$ is the problem dimension).
>
> It is true that in our methods we need to estimate the Lipschitz constant of the Hessian $L$. Note that it is commonly required to know the Lipschitz constants in theory for most of the established stochastic methods (both first-order and second-order algorithms). However, in our opinion, it is not a significant limitation. In some cases (e.g. Logistic Regression) we know the Lipschitz constant, or we can possibly efficiently estimate it. In general, we can consider it as a hyperparameter (a 'step size') which should be chosen using grid search. In our practical experience, second-order methods are typically much less sensitive to the exact value of the Lipschitz constant for the Hessian, as compared to the sensitivity of the first-order algorithms to the choice of step size. To the best of our knowledge, development of provable and efficient adaptive methods for stochastic second-order optimization is an interesting open question. We will add more discussion on that to our paper.
>
>
> 4. **Similarity measures.** The parameter $\delta_1$ (resp. $\delta_2$) measures how much the gradients (resp. Hessians) of the helper and the main function are aligned. In Assumption 3, these two parameters are the Hessian Lipschitzness of $f-h_i$, and so they can be estimated in practice by a difference of the Hessians or a difference between the gradient and its first-order approximation.
>
>
> **Minor questions:**
>
> 1. Thanks, we will fix it by introducing $M$ earlier.
>
> 2. $F_0$ is defined in line 94, it is simply $f(x_0) - f^\star$.
>
> 3. We use Lemmas 6 and 7 in the appendix.
>
> 4. Yes, it is a complexity guarantee, we can compare it to first order methods since we are talking in terms of GradCost (we express the cost of computing a Hessian as $d_{eff} \times$ that of computing a gradient).
>
> 5. The batches are not necessarily the same.
>
> 6. Thanks. Yes, we will link this.
>
> 7. This is a typo, we will correct it. Thanks.

---

> > ### Comment · Reviewer_bq1t · 2023-08-10
> >
> > Many thanks to the reviewers for their detailed response. I appreciate the table of global complexities and the discussion of the interpretation of $\delta_1$ and $\delta_2$ and how to set $m$. I raised the score I gave for the paper although I still have some reservations on (a) the practicality of the algorithm given the need to estimate $m$ (I agree that the literature generally assumes $L$ is known), (b) the usefulness of convergence guarantees that rely on additional variables $\delta_1$ and $\delta_2$.

---

> > > ### Author Response · Authors · 2023-08-14
> > >
> > > We would like to thank you again for your answer. We will try to address reservations and hope our answers are satisfying.
> > >
> > > - Regarding **(a)**, estimating $m$ is not problematic; it serves as a hyperparameter in our algorithm that we can determine using methods like grid search. In our article, we demonstrate that we can choose $m$ in a way that minimizes the total number of gradient calls. This number depends entirely on the knowledge of the count of gradients and Hessians utilized during each step. Surprisingly, this requirement only involves knowing the number of samples $n$ and the dimension $d$. To illustrate this, let's consider the case of the lazy Newton algorithm. In this scenario, we have $\delta_1=0$ and $\delta_2=L$, leading to $E[|\nabla f(x_{\text{out}})|^{3/2}]=\mathcal{O}( \frac{\sqrt{L}F_0}{S\sqrt{m}})$. To determine the optimal $m$ based on this rate, let's calculate the total number of gradient calls. It's evident that this count equals $n d S + n Sm = nS (d + m)$ because we used $S$ Hessians and $mS$ gradients. For a given precision $\varepsilon$, we have $S\sim \frac{\sqrt{L}F_0}{\varepsilon^{3/2}\sqrt{m}}$. Consequently, when minimizing $nS (d + m)$ with respect to $m$, we obtain $m=\sqrt{d}$. While this example is the simplest in the paper, the general idea is the same.
> > >
> > >
> > > - In response to **(b)**, to underscore the practical significance of our convergence guarantees hinging on parameters $\delta_1$ and $\delta_2$, we highlight instances (as discussed in the paper) where these parameters are precisely known. To be specific, Equation (10) in the paper presents the bound $\mathcal{O}\big(\frac{\sqrt{L}F_0}{Sm} + \frac{\sqrt{\delta_2}F_0}{S\sqrt{m}} + \frac{\sqrt{\delta_1}F_0}{S}\big)$, allowing for the direct substitution of known similarity parameter values. For example, in the case of cubic Newton's method, we have $\delta_1=\delta_2=0$. For Lazy-Newton, the values are $\delta_1=0$ and $\delta_2=L$. In the context of VRCN, $\delta_1=\mathcal{O}(\frac{L}{\sqrt{b_g}})$ and $\delta_2=\mathcal{O}(\frac{L}{\sqrt{b_h}})$, while for Lazy-VRCN, $\delta_1=\mathcal{O}(\frac{L}{\sqrt{b_g}})$ and $\delta_2=0$. This consolidated formulation encompasses all of these rates within a single formula.
> > >
> > > An insightful observation is that random batch sampling (performed in previous works), can be understood as an attempt to reduce similarity, and aligns with the objective of finding helpers with small similarities. However, it's worth noting that alternative methods might exist to achieve similar outcomes.

---

### Official Review · Reviewer_XQuJ · 2023-07-10

**Soundness:** 4 excellent
**Presentation:** 3 good
**Contribution:** 3 good
**Rating:** 6
**Confidence:** 4

**Summary:**

The paper provides a unified analysis for the Cubic regularized Newton Methods with variance-reduced estimators to solve stochastic nonconvex optimization problems. The results rely on the helper framework which generalizes the use of stochastic estimators for first-order and second-order oracles. The analysis recovers best-known rates and improves in terms of Hessian computation via Lazy Hessian update. Convergence is considered in general nonconvex and gradient-dominated (PL condition) problems.

**Strengths:**

The strengths of the paper lie within the unified convergence analysis that can recover best-known rates. This can be done via using the helper function which is a clever way to generalize the estimation of stochastic first-order amd second-order (possibly higher...) oracles.

Although Algorithm 1 is similar to the original Cubic Newton method but slightly difference in the gradient/Hessian estimators and the update of snapshot points.

All claims are backed by proof. I have gone through the proofs and I find no problems.

**Weaknesses:**

The paper lists 2 options to update the snapshot but there is lack of discussion on why these options are chosen (not others).

Other than that, I agree with the weaknesses pointed out by the authors, especially on the choice of the helper function. As helper function is a new concept, the paper does not provide much details on how to find a concrete choice of helper function given an application. Also note that details on the choice of helper function are not needed in the results due to the similarity assumption.

**Questions:**

As we have freedom to choose helper function, is there a preferred choice of helper function and what properties that we are looking for in general?

For the two options (4) and (5), the theoretical results only consider the first and I find nowhere else in the paper that uses the second option. Do experiments all use option (4)? If we use option (5) do we expect improvement in performance?

**Limitations:**

I believe the authors have clearly pointed out the limitations of this work.

---

> ### Author Rebuttal · Authors · 2023-08-09
>
> Thank you very much for the invested time and expertise, and for the positive feedback as well. We hope that all raised issues are properly addressed in our rebuttal.
>
> **Weaknesses**:
>
> - Yes, for the update of the snapshot we use options (4) and (5), the main reason is that we needed these two options in our proofs. In the general non-convex case we use option (4), and in the special case of gradient-dominated functions, we need option (5).
>
> **Questions:**
>
> - **Choice of the helper function:** In general it should be done on a case-by-case basis based on what we have as auxiliary information (unlabeled data, noisy data,...). We provide in the paper a few examples like in semi-supervised learning where the unlabeled data can be used to construct the helper function, this unlabeled data can be endowed with random labels, which is sufficient for linear and logistic regression, for other models we might need to use evolving labels (like in self-training for example). We also discussed the example of core-sets where a weighted subset of the dataset is chosen as an auxiliary (we can also use a random batch), we also mentioned auxiliary learning where this choice is much easier as you have auxiliary data besides the task that you are interested in solving.
>
>
> - **Properties that a helper function should have:** In our theory we need it to have a small similarity as we define it in Assumption 3, this means that in expectation their Hessians don’t change fast as you change the parameters.
>
>
> - **Snapshot update**: Section 4 needs option (5), we mentioned this (admittedly not clearly) in lines 149-151, we will mention this explicitly in Section 4. Yes, all experiments use option (4). We note that option (5) can be sometimes harder to compute, but should always be better than option (4).

---

### Official Review · Reviewer_zomT · 2023-07-22

**Soundness:** 3 good
**Presentation:** 3 good
**Contribution:** 2 fair
**Rating:** 3
**Confidence:** 4

**Summary:**

The paper studies stochastic Cubic Newton methods for solving possibly nonconvex minimization problems. The authors propose a flexible “helper framework” that unifies stochastic and variance-reduced second-order algorithms, offering global complexity guarantees.
This framework allows arbitrary batch sizes, noisy gradient and Hessian estimates, and includes variance reduction and lazy Hessian updates. The work achieves state-of-the-art complexities for stochastic and variance-reduced Cubic Newton methods without artificial logarithms. It also introduces a new efficient lazy stochastic second-order method for large dimension problems.

**Strengths:**

This paper proposes a new lazy stochastic second-order method.

**Weaknesses:**

- The paper lacks a discussion of variance-reduced methods in the literature, neglecting
numerous related papers on the subject.
- The experiments are only in Appendix and weak with a “toy” dataset.
- The loss function is just similar to “convex” although using the non-convex regularizer.
- Many assumptions have been used in the paper. That could restrict the applications of the
problems.
- It is not clear about the results that could use for machine learning applications.

**Questions:**

- What value of lambda for non-convex regularizer in line 411 did you choose?
- Is it more suitable to submit this paper to an optimization journal?

---

> ### Author Rebuttal · Authors · 2023-08-09
>
> Thank you very much for the invested time and expertise, and for the positive feedback as well. We hope that all raised issues are properly addressed in our rebuttal.
>
> **Addressing weaknesses:**
>
> 1. **Related works**. We did not discuss various variance-reduction first-order methods because the main focus of our paper is second-order optimization. We provided all related works on variance-reduced second-order methods that we are aware of [36, 30, 19]. We are happy to add more references. Please, find attached a PDF file with a new table with global complexities, which we also include in the main part of our paper. In this table, we compare our results with stochastic and variance-reduced first-order methods.
>
> 2. **Contributions and experiments**. We put the experiments in Appendix due to the lack of space. Note that the main results of our paper are theoretical, these are:
> - New second-order helper framework that covers multiple methods with global convergence guarantees in a unified way (stochastic second-order methods, variance reduction, Lazy methods, and semi-supervised learning). A technique similar in spirit to first-order methods was developed recently [8, 31].
> - Based on the helper framework, we equip the known variance-reduced methods with new complexity bounds for the important class of gradient-dominated functions, that include convex and strongly convex objectives.
> - We also propose a new stochastic second-order method (VRCN-Lazy) with improved global complexities (please, see also the table in the attached PDF).
> - In application to Auxiliary Learning, we show a provable benefit of using auxiliary tasks as helpers in our framework.
>
> Our illustrative numerical experiments are intended to support our theory and demonstrate that the methods indeed work in practice. Note that we use a real dataset in our experiments with the Logistic Regression model.
>
> 3. **Convexity of loss function**. Note that our theory covers both non-convex and convex objectives, with improved complexity guarantees in the latter case. We believe that convex optimization problems are also important for applications. We provide experiments for Logistic Regression with convex and non-convex regularizers. We followed the experimental setup of the predecessor work [18] on stochastic second-order methods, which we improved upon by adding the variance reduction and lazy Hessian computations. We also included experiments with a Diagonal Neural Network, which is non-convex.
>
> 4. **Assumptions**. Our assumptions are standard from the most related literature on second-order optimization methods. The main assumption about the objective function is that the Hessian is Lipschitz (Assumption 1), which is classic in the literature (see, e.g. [20]) and it is a natural generalization of the widely used first-order smoothness. Our Assumptions 2 and 3 do not restrict the problem class at all, but only specify the required properties of the gradient and Hessian approximations that we use in our helper framework (Algorithm 1). These assumptions are satisfied for helper functions that we use in our work, and simply demonstrate under which conditions we can ensure global convergence.
>
> 5. **Application to Machine Learning**. In general, stochasticity is a key feature of all prominent ML optimizers, thus motivating our study. We provided several use cases in the paper of how our framework can be applied in ML.  For more concrete examples, semi-supervised learning allows the use of unlabelled data to construct the helper function. We show that for semi-supervised linear and logistic regressions, we have $\delta_1 =\delta_2=0$.  Additionally, auxiliary learning has been gaining attention from the ML community. From the perspective of stochastic optimization, we thus think our work is relevant to ML.
>
> **Questions:**
> In both experiments, we took $\lambda = 1/n$ where $n$ is the number of samples.
>
> We believe that our results are well suited for the NeurIPS conference since the main focus of our theory is the development of stochastic second-order optimizers that have provably better complexity guarantees than first-order methods, and that can be efficiently implemented in practice.

---

### Author Rebuttal · Authors · 2023-08-09

A table comparing the convergence rates of different algorithms discussed in the paper.

---

### Decision · Program_Chairs · 2023-09-21

**Decision:**

Reject

**Comment:**

The paper studies stochastic Cubic Newton methods for solving possibly nonconvex minimization problems. Although the authors have provided the rebuttal, the reviewer finds that the theory is not impressive and the experiment is delayed to the appendix. They think more experiments are needed to demonstrate the efficiency of the theory. Moreover, the reviewer also pointed out that design helper function is quite application specific. It is not trivial to design helper function based on the assumptions. This paper is in borderline and has some potential. However, not all the reviewers support the publication of the paper. I strongly encourage the authors to consider the reviewers’ comments and suggestions and resubmit it to the next publication venues.